# Bioorthogonal labeling of transmembrane proteins with non-canonical amino acids unveils masked epitopes in live neurons

Diogo Bessa-Neto [1,5], Gerti Beliu [2,3,5], Alexander Kuhlemann [2,5], Valeria Pecoraro[1], Sören Doose [2], Natacha Retailleau[1], Nicolas Chevrier[1], David Perrais [1], Markus Sauer [2✉] & Daniel Choquet [1,4✉]

Progress in biological imaging is intrinsically linked to advances in labeling methods. The explosion in the development of high-resolution and super-resolution imaging calls for new approaches to label targets with small probes. These should allow to faithfully report the localization of the target within the imaging resolution – typically nowadays a few nanometers - and allow access to any epitope of the target, in the native cellular and tissue environment. We report here the development of a complete labeling and imaging pipeline using genetic code expansion and non-canonical amino acids in neurons that allows to fluorescently label masked epitopes in target transmembrane proteins in live neurons, both in dissociated culture and organotypic brain slices. This allows us to image the differential localization of two AMPA receptor (AMPAR) auxiliary subunits of the transmembrane AMPAR regulatory protein family in complex with their partner with a variety of methods including widefield, confocal, and *d*STORM super-resolution microscopy.

[1] University of Bordeaux, CNRS, Interdisciplinary Institute for Neuroscience, IINS, UMR 5297, F-33000 Bordeaux, France. [2] Department of Biotechnology and Biophysics, University of Würzburg, Biocenter, Am Hubland, 97074 Würzburg, Germany. [3] Rudolf Virchow Center for Integrative and Translational Bioimaging, University of Wuerzburg, Wuerzburg, Germany. [4] University of Bordeaux, CNRS, INSERM, Bordeaux Imaging Center, BIC, UMS 3420, US 4, F-33000 Bordeaux, France. [5]These authors contributed equally: Diogo Bessa-Neto, Gerti Beliu, Alexander Kuhlemann. ✉email: m.sauer@uni-wuerzburg.de; daniel.choquet@u-bordeaux.fr

Over the past 15 years, advances in light-based super-resolution microscopy have revolutionized the way neuroscientists perceive key neuronal processes such as synaptic and axonal nanoscale organization or protein trafficking at the single-molecule level[1–3]. The improvements in the various super-resolution imaging methods, and particularly in single-molecule localization microscopy (SMLM), have made it possible to routinely reach imaging resolutions in the order of ~20 nanometers. However, elucidating target protein organization at virtually molecular resolution requires not only a high localization precision of individual emitters but also a high labeling density and specificity, and a distance between the fluorescent reporter and the target (linkage error) substantially smaller than the desired imaging resolution[4,5]. Classical labeling methods used for fluorescence imaging such as labeling the target protein with an antibody-dye complex or genetic fusion with a reporter fluorescent protein are limited in their use, particularly in live neurons. Antibodies are quite bulky, even when reduced to their monovalent forms, and only have access to exposed epitopes. Incorporation of fluorescent protein to a target protein can impede its native function, accounting for biased interpretations and severely limiting its possible site of insertion. There is thus a pressing need for the development of alternative labeling methods that do not depend on epitope accessibility and with sizes compatible with the nanometer precision of super-resolution imaging.

The nanoscale organization of synapses is an ideal model system for the application of innovative imaging and labeling methods because of its exquisite complexity and diversity as well as because of the tight link between synapse dynamic organization and function[2]. Among synaptic proteins, the complex involved in regulation of the function, localization, and trafficking of AMPA receptors (AMPAR) – the glutamate receptors that mediate most excitatory synaptic transmission, has historically raised large interest. Transmembrane AMPAR regulatory protein (TARP) family are four transmembrane proteins characterized by an intracellular amino- and carboxyl-terminal domain, and two extracellular loops (Ex1 and Ex2)[6] (Supplementary Fig. 1a). TARPs are key modulators of AMPAR-mediated synaptic transmission and plasticity, as they promote AMPAR surface targeting, regulate their pharmacology and gating which are fundamental for proper AMPAR-mediated transmission[7–13]. Among the different members of the TARP family, γ2 (also known as stargazin) is the prototypical AMPAR auxiliary subunit and has been the most widely studied, followed, more recently, by γ8 that is the most abundant TARP in the hippocampus[14]. While sharing a large homology[15], γ2 and γ8 not only exert a differential modulation of AMPAR[13,16,17] but also display differential plasma membrane distribution, with γ2 suggested to bear an almost exclusive synaptic localization and γ8 a more widespread dendritic distribution, as seen in electron microscopy studies[18,19]. This was never, to the best of our knowledge, confirmed in living neurons by optical microscopy due to the lack of adequate tools.

Our understanding of TARPs localization and trafficking has been hampered by a lack of suitable labeling and imaging tools. The close association of the extracellular domains of TARPs to the AMPAR ligand-binding domain (LBD)[20–24], that confer their role in TARP-specific AMPAR modulation[13,16,25,26], has hindered the development of ligands recognizing the extracellular domains of TARPs as well as genetic fusion tagging[27,28]. Deciphering the respective surface diffusion and synaptic organization properties of γ2 and γ8 at the nanoscale is of particular interest given their presumptive key role in AMPAR regulation as well as in the control of synaptic plasticity, but their molecular organization unfortunately remains widely unknown. This question is becoming particularly relevant given the recent increased interest in their differential role in synapse organization and function[24,29].

Click chemistry labeling via genetic code expansion (GCE) offers the possibility for site-specific incorporation of non-canonical amino acids (ncAAs) containing bioorthogonal groups into a target protein[30,31]. By replacing a native codon at a selected position in the target protein with a rare codon, such as the Amber (TAG) stop codon, the modified protein can then be expressed in the desired host cells along with an engineered aminoacyl-tRNA synthetase (aaRS) and tRNA pair orthogonal to the host translational machinery. The engineered aaRS is modified in a way to only recognize a specific ncAA, which is then attached to a tRNA that matches the rare codon. Among different possibilities, the trans-cyclooct-2-ene (TCO*)-modified amino acids, such as TCO*-L-lysine (termed TCO*-A, where A stands for Axial isomer), is of interest when it comes to targeting and labeling the desired target proteins in living organisms. TCO* can react with a 1,2,4,5-tetrazine in a catalyst-free, fast, specific, and bioorthogonal strain-promoted inverse electron-demand Diels-Alder cycloaddition reaction (SPIEDAC). Due to the high selectivity and fast kinetics of this click chemical reaction, a large number of fluorophore-tetrazine conjugates and TCO*-functionalized molecules are now commercially available, making labeling of mammalian cells and whole organisms with organic dyes accessible for live and fixed samples[32–34].

Here, we explore the potential of bioorthogonal labeling as a strategy to tag and visualize surface TARPs in living neurons by conventional and super-resolution microscopy with minimal to non-perturbation of TARPs modulation of AMPAR, opening doors to the study of hard-to-tag proteins in living neurons. We describe a complete pipeline that allows labeling proteins in live neurons in both primary and organotypic hippocampal cultures by GCE. Using this approach, we report the differential subcellular distribution of γ2 and γ8 at the light microscopy and single-molecule level. Also, our antibodies directed against extracellular loops of γ2 and γ8 allow us to establish that there are virtually no free surface γ2 and γ8 in hippocampal neurons.

## Results

**Epitope masking by close interaction of TARPs extracellular loops with AMPAR LBD.** As a first attempt to create specific ligands for γ2 and γ8 that could be used to study their organization and trafficking in live neurons, we generated antibodies against the γ2 Ex2 and γ8 Ex1 (Supplementary Fig. 1b). We first evaluated the antibodies specificity by incubating living COS-7 cells expressing either γ2 or γ8 bearing mEos2 as a reporter. As shown in Supplementary Fig. 1c, both antibodies are specific towards their respective target protein. We then analyzed if we could use these antibodies to label endogenous TARPs in dissociated hippocampal neurons, as both γ2 and γ8 are expressed in the hippocampus[15,35,36]. To our surprise, our antibodies were unable to recognize endogenous γ2 or γ8 in our primary hippocampal cultures. The presence of γ8 was confirmed by post-fixation immunostaining against the intracellular C-terminal domain of γ8 (Supplementary Fig. 1d), whereas γ2 immunostaining was inconclusive due to the poor sensitivity of the tested commercial α-γ2 antibodies. To understand the lack of TARPs staining in neurons, we overexpressed in neurons γ2 fused to eGFP at the C-terminus (γ2::eGFP) or GluA2 tethered to γ2::eGFP (GluA2::γ2::eGFP) in which the GluA2 C-terminus is fused to the γ2 N-terminus by in-frame expression[37]. When labeled with the α-γ2 Ex2, neurons overexpressing γ2::eGFP displayed specific antibody labeling that colocalized with the eGFP signal, likely revealing AMPAR-free γ2::eGFP. In contrast,

in neurons overexpressing GluA2::γ2::eGFP and labeled with the α-γ2 Ex2, no antibody labeling was observed (Supplementary Fig. 1e). Structural data of γ2 or γ8 in complex with AMPARs revealed the close proximity of both Ex1 and Ex2 to the AMPAR ligand-binding domain (LBD)[21–23], which likely results in epitope masking. This finding leads to the conclusion that antibodies fail to recognize endogenous TARPs in dissociated hippocampal neurons as well as in GluA2::γ2::eGFP-overexpressing neurons because of steric inaccessibility. Altogether, our results indicate that at the plasma membrane endogenous TARPs are always associated with AMPARs that mask the extracellular epitopes.

**Genetic code expansion and bioorthogonal labeling of TARPs.** Bioorthogonal labeling of proteins by replacing a single natural amino acid with a ncAA has emerged in the past years as an alternative strategy to target and visualize proteins in living mammalian cells with minimal to no perturbation[32,33,38]. Because click chemistry labeling of ncAAs with tetrazine-dyes is efficient and sterically minimally demanding, we hypothesized that it might be the method of choice to label masked epitopes on TARPs. We first designed Amber mutants (herein termed ncAA-tagged) of γ2 and γ8 by site-direct mutagenesis (Fig. 1a), with respect to previously conducted work on γ2 with ncAAs[20]. Additionally, we replaced the endogenous Amber termination codon of γ8 with the Ochre codon (TAA) to prevent erroneous ncAA incorporation at the C-terminus (see Methods section). To identify the best position for the insertion of the Amber codon, three mutants of γ2 (S44*, S51*, and S61*) and γ8 (S72*, S84*, and K102*) were tested for ncAA incorporation and labeling efficiency in HEK293T cells (Fig. 1b–d and Supplementary Fig. 1f, g).To check the efficiency of ncAA incorporation in the different mutants, eGFP was fused within the C-tail of γ2 and γ8–downstream the Amber mutation. Hence, inefficient incorporation or absence of ncAAs would result in premature translation termination and loss of eGFP signal (Fig. 1c, d). HEK293T cells were co-transfected with one of the different ncAA-tagged TARPs, and an engineered pyrrolysine aminoacyl tRNA synthetase (PylRS) and its cognate tRNA: single-copy tRNA[Pyl] (tRNA[Pyl])[39] or four copies (4xtRNA[Pyl])[40]. The clickable trans-cyclooctene derivatized lysine (TCO*A) was added to the cell media at the time of transfection for a period of ~24 h. Surface expression of the ncAA-tagged TARPs was accessed by bioorthogonal labeling via SPIEDAC using cell-impermeable tetrazine-dyes (H-Tet-Cy3, H-Tet-Cy5, and Pyr-Tet-ATTO643, Supplementary Fig. 2); tetrazines and TCO* react in a catalysis-free 'click-reaction'[32,33,41]. Afterwards, excess of tetrazine-dyes was removed by subsequent washes and cells were live-imaged using confocal microscopy. As indicated by the eGFP signal, all ncAA-tagged TARPs showed equivalent expression levels as compared to γ2::eGFP, revealing efficient incorporation of TCO*A. However, the mutant γ8 K102*::eGFP displayed decreased Pyr-Tet-ATTO643 labeling as compared to mutants γ8 S72*::eGFP and γ8 S84*::eGFP. No noticeable difference in tetrazine-dye labeling efficiency was observed within the different γ2 mutants (Supplementary Fig. 1f, g). Additionally, tetrazine-dye labeling was entirely due to the incorporation of TCO*A as no labeling was detected in cells transfected with γ2::eGFP in the presence of TCO*A nor in cells transfected with γ2 S44*::eGFP or γ8 S72*::eGFP in the absence of TCO*A (Fig. 1c, d and Supplementary Fig. 1f).

**ncAA-tagged TARPs physically and functionally interact with AMPAR-subunit GluA1 as seen by FRET and electrophysiology.** TARPs are auxiliary subunits to AMPARs that bind and interact closely with GluA subunits, as demonstrated by biochemical[11,27], functional[11], and structural[21–23] data. We thus aimed to study whether ncAA-tagged TARPs could still physically and functionally interact with GluA subunits using both Förster

Resonance Energy Transfer (FRET) measurements and electrophysiology. As we were able to insert ncAAs into the Ex1 loops of γ2 and γ8 that are in close apposition to the extracellular domain of GluA subunits[21–23], we first sought to use FRET to measure AMPAR-TARP interactions. We designed a set of possible FRET pairs between the ncAA-tagged γ2/γ8 and the AMPAR-subunit GluA1. To label surface GluA1, a SNAP-tag was either inserted at the N-terminus (SNAP::GluA1; no FRET expected) or within the ATD-LBD linker of GluA1 at position 396 aa (GluA1::SNAP396; potential FRET pair) (Fig. 2a). HEK293T cells were co-transfected with PylRS/tRNA[Pyl], SNAP-tagged GluA1, and the various ncAA-tagged TARPs in the presence of TCO*A. Cells were stained with 5 μM BG-AF488 (donor) and 1.5 μM H-Tet-Cy3 (acceptor) at 37 °C for 30 min. Excess dye was removed by subsequent washing steps with HBSS. Fluorescence lifetime imaging microscopy (FLIM) was used to estimate the degree of FRET-based changes of the donor AF488 fluorescence lifetime[42]. As expected, when co-expressed with the SNAP::GluA1-AF488, neither γ2 S44*-Cy3 nor γ8 S72*-Cy3 were able to quench the donor, i.e., no decrease in AF488 fluorescence lifetime was observed as compared to the donor alone condition (SNAP::GluA1-AF488 + γ2 S44*-Cy3: $\tau_{AF488}$ = 3.07 ± 0.03 ns, SNAP::GluA1-AF488 + γ8 S72*-Cy3: $\tau_{AF488}$ = 3.04 ± 0.01 ns, as compared to donor alone: no H-Tet-Cy3, GluA1::SNAP396-AF488 + γ2 S44*: $\tau_{AF488}$ = 3.01 ± 0.01 ns).

In contrast, we observed a robust decrease in GluA1::SNAP396-AF488 fluorescence lifetime when co-expressed with γ2 S44*-Cy3 or γ2 S61*-Cy3 as compared to the SNAP::GluA1-AF488 + γ2 S44*-Cy3, with the FRET pair GluA1::SNAP396-AF488 + γ2 S44*-Cy3 showing a stronger reduction in AF488 lifetime (Fig. 2b, c). Moreover, when we forced a one to one interaction between GluA1 and γ2 using a tethered GluA1 SNAP396 to γ2 S61* (GluA1::SNAP396::γ2 S61*-AF488/Cy3), we did not observe a significant difference in AF488 lifetime compared to the condition in which we expressed the two proteins separately (GluA1::SNAP396-AF488 + γ2 S61*-Cy3). This suggests a full occupancy of the AMPAR subunits with four TARPs under our experimental conditions. Similar to ncAA-tagged γ2-Cy3, the presence of ncAA-tagged γ8-Cy3 led to a robust decrease in GluA1::SNAP396-AF488 lifetime, with the GluA1::SNAP396-AF488 + γ8 S84*-Cy3 pair outperforming the pairs GluA1::SNAP396-AF488 + γ8 S72*- and γ8 K102*-Cy3 (Fig. 2d). As for γ2, we did not observe a difference between the GluA1::SNAP396-AF488 + γ8 K102*-Cy3 and the tethered GluA1 SNAP396 to γ8 K102* (GluA1::SNAP396::γ8 K102*-AF488/Cy3). Altogether, our FRET experiments indicate that ncAA-tagged TARPs physically interact with AMPAR and provide thus a tool to study the regulation of AMPAR-TARP interactions.

TARPs type-I, including γ2 and γ8, modulate AMPAR gating in a TARP subtype-specific manner[13,16,17,25,43]. To determine if the incorporation of TCO*A within the Ex1 loop compromises TARP function, notably interaction with and modulation of AMPARs, we performed whole-cell patch-clamp recordings in HEK293T cells co-expressing GluA1 (flip isoform) alone (eGFP, control) or in the presence of WT or ncAA-tagged γ2/γ8, bearing eGFP as a reporter. When compared to GluA1 alone and GluA1 in the presence of WT γ2 or γ8, the incorporation of TCO*A into the Ex1 loop did not impair the ability of TARPs to increase the efficacy of the partial agonist kainate (KA) over GluA1 (ratio peak current amplitude KA/Glu mean ± SD: GluA1 = 0.02 ± 0.01; GluA1 + γ2 = 0.80 ± 0.06; GluA1 + γ2 S44* = 0.80 ± 0.10; GluA1 + γ8 = 0.37 ± 0.11; GluA1 + γ8 S72* = 0.28 ± 0.09) (Fig. 2e, f). Additionally, we did not observe perturbations on TARPs ability to decrease receptor desensitization ($\tau_{des}$: GluA1 = 4.34 ± 0.59; GluA1 + γ2 = 8.44 ± 1.50; GluA1 + γ2 S44* = 9.02 ± 2.14; GluA1 + γ8 = 17.41 ± 3.72; GluA1 + γ8 S72* = 15.43 ± 2.82 ms; Supplementary Fig. 3a) or increase receptor recovery from desensitization ($\tau_{rec}$: GluA1 = 162.6; GluA1 + γ2 = 57.1; GluA1 + γ2 S44* = 62.3; GluA1 + γ8 = 40.9; GluA1 + γ8

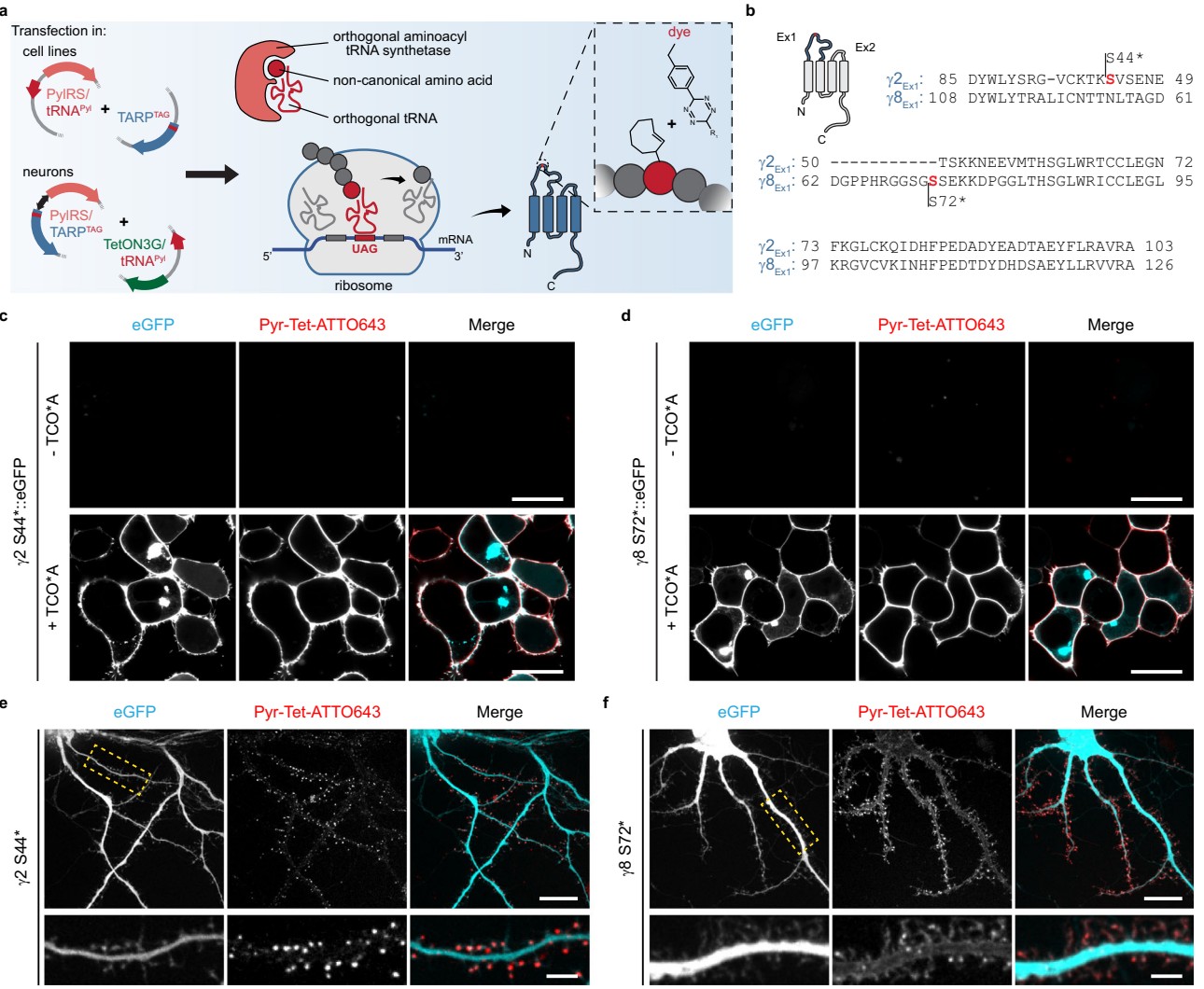

**Fig. 1 Bioorthogonal labeling of TARPs. a** Schematic overview of click chemistry labeling via genetic code expansion. Amber mutants of ncAA-tagged γ2 and γ8 were designed by standard site-direct mutagenesis to incorporate the ncAAs carrying a TCO*A for click labeling. Protein expression occurs through endogenous and orthogonal tRNA-synthetases in the presence of appropriate tRNAs. Labeling of ncAA-tagged proteins occurs through a catalysis-free reaction between TCO*A and tetrazine-functionalized dyes via the strain-promoted inverse electron-demanding Diels-Alder cycloaddition (SPIEDAC) reaction. (**b**) Sequence alignment of the first extracellular loops of γ2 and γ8 from *rattus norvegicus*. Amber substitution mutations are represented in red. **c**, **d** Representative confocal images of living HEK293T cells co-expressing PylRS/4xtRNA$^{Pyl}$ and **c** γ2 S44*::eGFP or **d** γ8 S72*::eGFP in the absence (upper) or presence of 250 μM TCO*A (lower) stained with 1.5 μM Pyr-Tet-ATTO643. Scale bar: 20 μm. Images are representative of three independent experiments. **e**, **f** Representative spinning disk confocal images of living dissociated hippocampal neurons co-expressing eGFP, Tet3G/tRNA$^{Pyl}$ and **e** pTRE3G-BI PylRS/γ2 S44* or **f** pTRE3G-BI PylRS/γ8 S72* in the presence of 250 μM TCO*A and 100 ng mL$^{-1}$ doxycycline labeled with 0.5 μM Pyr-Tet-ATTO643. Bottom panels, magnified views of segments of dendrites highlighted in the eGFP panel (dashed yellow boxes) of the overview images showing the distribution of γ2 S44* and γ8 S72*. Scale bar: 20 μm (overview images) and 5 μm (magnified images). All example images are representative of at least three independent preparations.

S72* = 45.8 ms; Fig. 2g). Furthermore, the presence of endogenous intracellular polyamines leads to a block of calcium-permeable AMPARs, like GluA1 homomers; better illustrated by a strong inwardly rectifying I-V curve. TARPs attenuate polyamine block of calcium-permeable AMPARs reducing AMPAR rectification[44]. No difference was observed between WT TARP and respective ncAA-tagged TARP ability to reduce the GluA1 rectification (Supplementary Fig. 3b). Altogether, the patch-clamp experiments indicate that ncAA-tagged TARPs retain a normal functional ability to modulate AMPAR gating.

**Distinct surface distributions of ncAA-tagged γ2 and γ8 in hippocampal neurons.** After verification of the proper ncAA incorporation into TARPs extracellular loop and the normal

receptor function in heterologous cells, we exploited the use of GCE to label recombinant TARPs in living neurons. The occurrence of the Amber codon in mammalian cells is rare, (~0.5 ‰), and represents 20–23% of all stop codons. It is however important to keep in mind that GCE might induce toxicity due to tRNA suppression of endogenous proteins containing Amber codon terminations. A good practice is to restrict the concentration of PylRS to prevent suppression of naturally occurring Amber codon terminations, as in the gene *Cacng8*, encoding the protein γ8. In addition, the presence of TARPs, especially γ8, increases AMPAR Glu affinity as well as channel conductance[26,45], as a result, overexpression of TARPs triggers Glu-induced excitotoxicity[46]. Hence, we decided to overexpress the PylRS and ncAA-tagged TARPs under a bidirectional doxycycline-

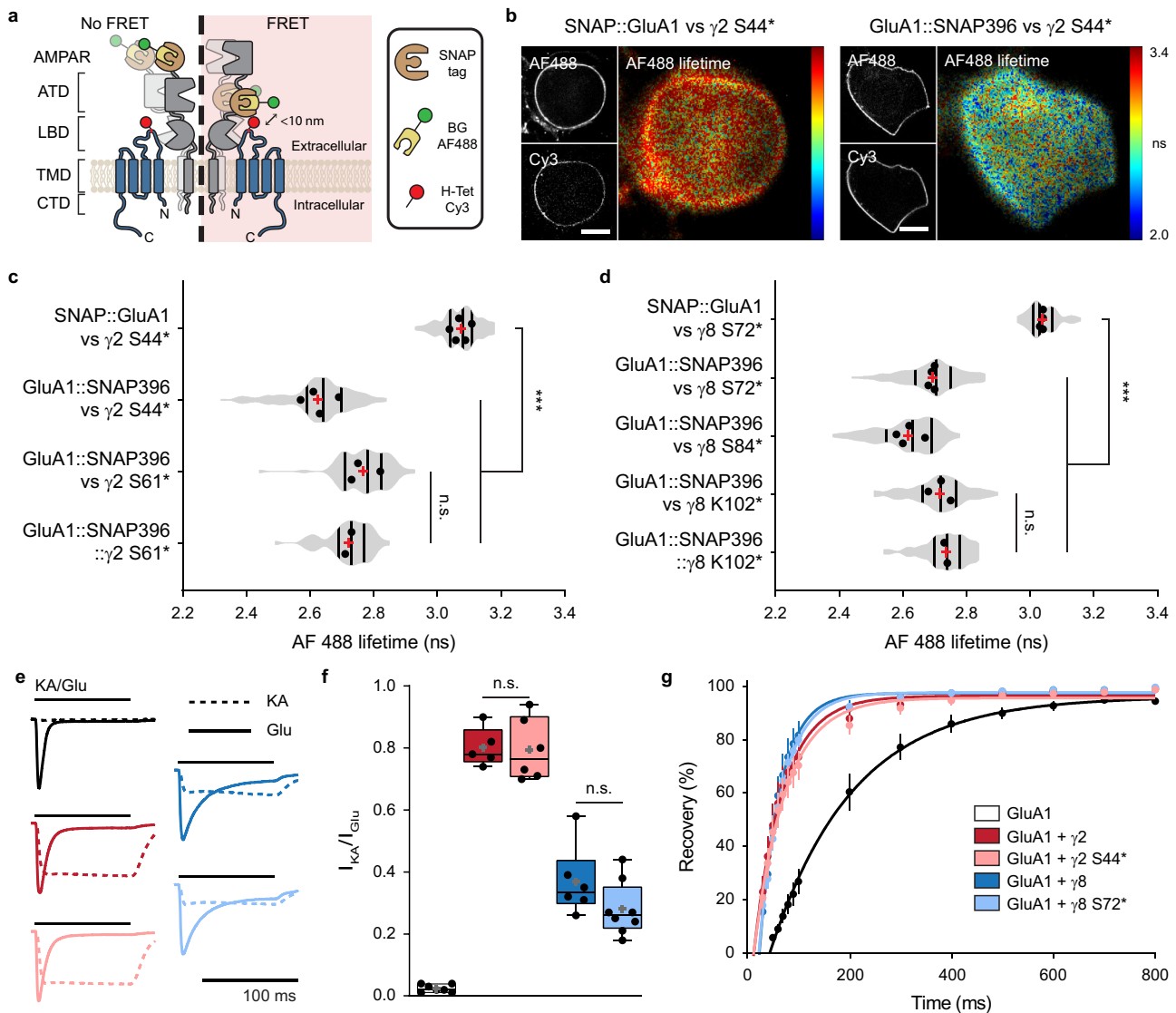

**Fig. 2 ncAA incorporation in the Ex1 loop of TARPs does not impair physical or functional interaction with AMPAR-subunit GluA1. a** Strategy to label GluA1 and TARPs for FRET microscopy. SNAP-tag was inserted at GluA1 N-terminus (SNAP::GluA1) or ATD-LBD linker (GluA1::SNAP396) and labeled with BG-AF488 (donor). ncAA-tagged TARPs were labeled with tetrazine Cy3 (acceptor). **b** Representative spinning disk (AF488 and Cy3) and widefield illumination FLIM (AF488 lifetime) images of living HEK293T cells co-expressing PylRS/tRNA$^{Pyl}$, γ2 S44*, and SNAP GluA1 (left) or GluA1 SNAP396 (right) used in **c**. Scale bar: 10 μm. **c, d** Average AF488 lifetime in HEK293T cells co-expressing PylRS/tRNA$^{Pyl}$, SNAP-tagged GluA1 (SNAP::GluA1 or GluA1::SNAP396), and **c** ncAA-tagged γ2 (γ2 S44* or γ2 S61*) or **d** ncAA-tagged γ8 (γ8 S72*, γ8 S84*, or γ8 k102*), or alternatively PylRS/tRNA$^{Pyl}$ and **c** GluA1::SNAP396::γ2 S61* or **d** GluA1::SNAP396::γ8 K102*. From top-to-bottom, cell numbers were **c** 117, 108, 66, 51; **d** 87, 104, 111, 84, 53. Data was pulled from two-to-five independent preparations. Statistical significance was analyzed using Welch's ANOVA test; ***$p < 0.001$, n.s. specifies no significance. **e** Whole-cell currents and **f** ratios of KA- to Glu-evoked currents in response to 0.1 mM KA (dashed) or 10 mM Glu (line), or **g** recovery of desensitization to two pulses of 100 ms Glu applied at different intervals from HEK293T cells co-transfected with PylRS/tRNA$^{Pyl}$, GluA1, and eGFP (control, black, **f** 7 and **g** 6 cells), γ2::eGFP (dark red, 5 cells), γ2 S44*::eGFP (light red, 6 cells), γ8::eGFP (dark blue, 6 cells), or γ8 S72*::eGFP (light blue, 8 cells) from three-to-four independent preparations. Statistical difference was analyzed using one-way ANOVA with a Fisher's LSD multiple comparisons test; n.s. specifies no significance. All data represent mean ± SD. Box and violin indicates 25th to 75th percentiles, with median represented as a centre line, mean represented as a cross, and on box plot: whiskers represent max to min. Source data are provided as a Source Data file.

inducible promoter, pTRE3G-BI, i.e. pTRE3G-BI PylRS/γ2 S44* and pTRE3G-BI PylRS/γ8 S72*. To make it easier to follow, we will simply refer to these constructs by the name of the respective ncAA-tagged TARP (γ2 S44* or γ8 S72*), however it should be noted that different DNA constructs were used for the expression of ncAA-tagged TARPs in heterologous cells and neurons. To further decrease the complexity of our tool, we combined in a single vector the tRNA$^{Pyl}$ and the Tet-On 3G transactivator (herein termed Tet3G/tRNA$^{Pyl}$), which binds to and activates

expression from TRE3G promoters in the presence of doxycycline (see Methods section).

To estimate the potential off-target surface labeling level in our neuronal experiments, dissociated hippocampal neurons at days in vitro (DIV) 3–4 were co-transfected with the Tet3G/tRNA$^{Pyl}$, pTRE3G-BI PylRS (no TARP, only the PylRS), and clickable-GFP. Five days before H-Tet-Cy5 labeling, expression of PylRS was induced by doxycycline, and TCO*A was added to the cell media. Approximately 24 h before labeling, half of the cell media

was replaced by fresh media supplemented with doxycycline and TCO*A (see Methods section). As shown in Supplementary Fig. 4, expression of clickable-GFP indicates the success of the GCE experiment, while the transfected cell does not express more H-Tet-Cy5 labeling than non-transfected neighboring neurons, demonstrating the absence of any detectable off-target surface labeling in the absence of clickable surface proteins.

To express TARPs, dissociated hippocampal neurons were transfected with the necessary machinery for the expression of γ2 S44* or γ8 S72* together with eGFP. At DIV 16-17, 100 ng mL$^{-1}$ doxycycline and 250 μM TCO*A were added to the cell media for ~20 h. Similar to bioorthogonal labeling of HEK293T cells, surface labeling of ncAA-tagged TARPs was obtained by live incubation with 0.5 μM of cell-impermeable tetrazine-dyes (H-Tet-Cy5, Pyr-Tet-ATTO643, Pyr-Tet-AF647). Excess of tetrazine-dye was removed by subsequent washes with warm Tyrode's solution before imaging of live or fixed neurons (Figs. 1e, f and 3). Similar to what we observed in HEK293T cells, TCO*A-supplemented neurons transfected with either WT γ2 or WT γ8 displayed no tetrazine-dye labeling (Fig. 3a–c upper panel and d–f upper panel, respectively). Surface labeling of γ2 S44*-positive neurons showed a strong enrichment of γ2 S44* in the dendritic spines with low expression in the dendritic shaft (Figs. 1e and 3a–c lower panel). In contrast, γ8 S72* was distributed throughout the dendritic arbor (Fig. 1f and Fig. 3d–f lower panel). Of note, in Fig. 1e, f, only surface TARPs are revealed, contrary to the images of GFP in neurons expressing γ2::GFP (Supplementary Fig. 1e) that display both, surface and intracellular γ2::GFP, explaining the more diffuse labeling in the latter case.

Using eGFP as a cell marker, we compared the enrichment of γ2 S44* and γ8 S72* in spines versus dendritic shafts, at the base of the measured spine (extraspine). We observed an average of 2.60 ± 0.69 fold higher enrichment in the spines as compared to the neighboring shaft for γ2 S44* (Fig. 3g). Furthermore, γ2 S44* showed a pronounced tendency to accumulate in clusters heterogeneously distributed in the spine head. In contrast, γ8 S72* had a more homogeneous distribution in dendrites and spines with a lower tendency to form clusters. When comparing the fluorescence levels in the spine to the neighboring extraspine area, γ8 S72* was only slightly enriched at the spines (1.17 ± 0.25 fold increase; Fig. 3g).

One of the drawbacks of overexpression systems, in particular transfection, is the heterogeneous expression levels of the protein of interest from cell to cell and the potential higher expression level as compared to the endogenous protein, which might lead to artifacts like mislocalization of proteins. We directly estimated the overexpression level of γ8 by immunocytochemistry based on the total γ8 levels in γ8 S72*-positive neurons compared to non-transfected neurons (Supplementary Fig. 5). Our data showed that, while some neurons display up to 3–4-fold higher γ8 expression levels than the average, the mean γ8 expression level in all transfected neurons was not significantly different from that in non-transfected neighboring neurons. We could not perform this control for γ2 levels due to the poor quality of the staining we obtained with the C-terminal γ2 antibody. Interestingly, this might be related to poor accessibility of the antibody to the γ2 C-terminus in the packed PSD, as was previously observed in TEM[19]. This effect would be less prominent for γ8 that is more extrasynaptic. As a note, the fact that we observed strong synaptic localization of γ2 by GCE labeling confirms that our observation does not result from overexpression-induced γ2 mislocalization. Indeed, overexpression would tend to saturate γ2 binding sites in the PSD and lead to more extrasynaptic γ2.

To further explore the impact of γ2 and γ8 expression levels on their distribution, we plotted the mean fluorescence intensity level measured on all analyzed extraspine areas versus the neighboring spine levels. We observed a poor correlation between γ2 S44* labeling at spines versus extraspine, while γ8 S72* displayed a strong correlation between these two areas, independently of the expression level (Fig. 3h). Furthermore, the average enrichment ratio per neuron of both γ2 S44* and γ8 S72* was independent of the expression level (Fig. 3i). This indicates that the difference observed between γ2 S44* and γ8 S72* distribution is independent of their expression level, and likely due to the intrinsic nature of the proteins and their targeting properties.

### Bioorthogonal labeling of TARPs in organotypic hippocampal slice cultures.
While dissociated primary neuronal cultures are a well-established experimental model, they lack the physiological cellular environment, network, and regional specificity of the intact brain. Given the small size of tetrazine-dyes, high specificity, and ultrafast bioorthogonal reaction with TCO*, we aimed to exploit the potential of this approach as a tool to label surface proteins in the more physiological system of organotypic hippocampal slice cultures (OHSC). We used single-cell electroporation (SCE) to deliver the cDNAs in identified target CA1 pyramidal neurons from 300 μm thick slices. Similar to what we achieved in dissociated neurons, we used the doxycycline-inducible expression system for the controlled expression of ncAA-tagged TARPs and PylRS. TCO*A and doxycycline were added to the media approximately 22 h before tetrazine-dye labeling. Excess of TCO*A and tetrazine-dye in the extracellular space were removed by subsequent washes with warm Tyrode's solution (Fig. 4a, see Methods section). Confocal images of fixed slices of SCE CA1 neurons co-expressing eGFP and γ2 S44* or γ8 S72* demonstrated good tissue penetrability and high specificity of H-Tet-Cy5 towards TCO* for tissue applications as indicated by the eGFP signal (Fig. 4b, c, e). Similar to what we observed in dissociated neurons (Fig. 3a–f lower panel), γ2 S44* expressed into CA1 neurons showed a remarkable fluorescence signal enrichment at spines of both apical and basal dendrites with reduced labeling in the dendritic shaft (Fig. 4d, g, h). To verify γ2 S44* accumulation along the Z-projected dendritic shaft (Fig. 4d), we co-expressed γ2 S44* with the PSD-95 marker XPH20 fused with eGFP (XPH20::eGFP)[47,48] as a reporter and found that γ2 S44* accumulation was indeed always colocalized with the XPH20 eGFP signal (Fig. 4k). In contrary to γ2 S44*, but in line with the observations made in dissociated neurons overexpressing γ8 S72* (Fig. 3d–f lower panel), γ8 S72*-overexpressing CA1 neurons showed a more homogeneously distributed H-Tet-Cy5 fluorescence signal along the dendrites (Fig. 4f, i, j).

This highlights the reliability of bioorthogonal labeling as a versatile, fast, and specific tool for live labeling of proteins in neuronal tissue.

### dSTORM imaging reveals differences in nanoscale distribution of TARPs.
To investigate the peculiar difference found in the distribution of γ2 S44* and γ8 S72* in neurons by confocal microscopy in more detail, we used SMLM by *direct* stochastic optical reconstruction microscopy (dSTORM)[49,50]. dSTORM images revealed the molecular distribution of γ2 S44* and γ8 S72* in hippocampal neurons (Fig. 5a, b) and indicated that γ2 S44* accumulates in synaptic spines (Fig. 5b–d), in agreement with the confocal data. To quantify the distribution of ncAA-tagged and H-Tet-Cy5 clicked TARPs, we co-expressed again the PSD-95 marker XPH20::eGFP as a reporter to identify synaptic sites and compared the localization densities determined from dSTORM data of extrasynaptic and synaptic sites. While both TARPs show a homogeneous distribution in extrasynaptic sites, the absolute localization density determined for γ8 S72* is ~3-

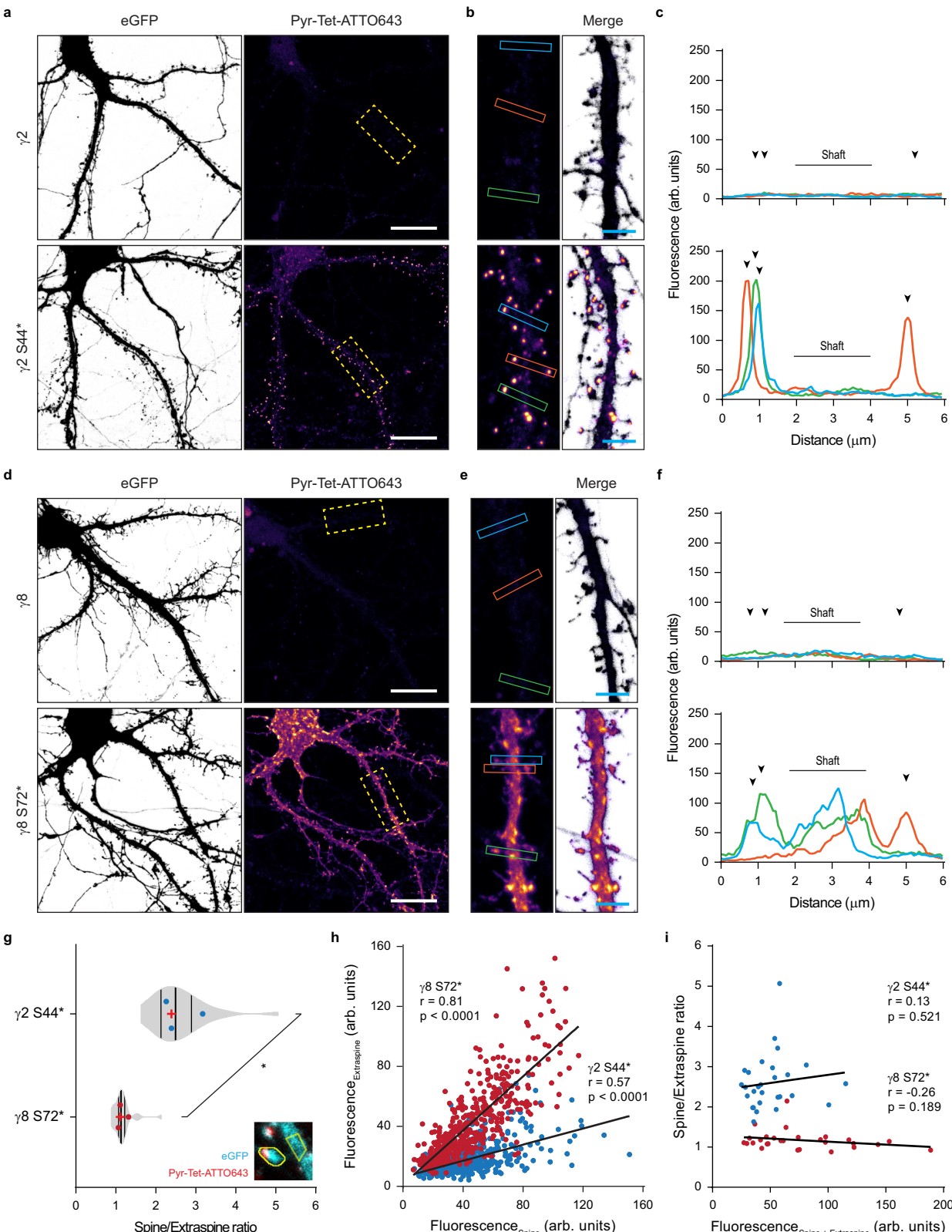

fold higher (Fig. 5c). Together with the slightly higher localization density of γ2 S44* in synaptic sites (Fig. 5c), our data thus demonstrate that the localization density measured for γ2 S44* is ~9 fold higher in synaptic as compared to extrasynaptic sites, whereas γ8 S72* exhibits only a ~2 fold higher localization density in synaptic compared to extrasynaptic sites (Fig. 5c, inset).

Next, we calculated Ripley's K-function of several regions of interest (ROIs) in synaptic and extrasynaptic areas to analyze the distribution of TARPs in neurons (Supplementary Fig. 6) and compared them to simulated data with spatial distributions following complete spatial randomness or a clustered Neyman–Scott process (accounting for multiple localizations from each fluorophore) in identical ROIs. Ripley's K-functions showed for

**Fig. 3 Distinct dendritic surface distribution of γ2 S44\* and γ8 S72\* in dissociated neurons. a–d** Representative confocal images of fixed dissociated hippocampal neurons co-expressing eGFP, Tet3G/tRNA$^{Pyl}$ and **a** pTRE3G-BI PylRS/γ2 (upper panel), pTRE3G-BI PylRS/γ2 S44\* (lower panel), **d** pTRE3G-BI PylRS/γ8 (upper panel), or pTRE3G-BI PylRS/γ8 S72\* (lower panel) in the presence of 250 μM TCO\*A and 100 ng mL$^{-1}$ doxycycline live stained with 0.5 μM Pyr-Tet-ATTO643. **b, d** 20 μm and **b, e** 4 μm (magnified images). **c, f** Line scan measurements of Pyr-Tet-ATTO643 across spines and dendritic shaft based on eGFP signal represented in **b** and **e**. **g** Average spine to extraspine intensity ratio of the ncAA staining indicating a spine enrichment of 2.60 ± 0.69 folds for γ2 S44\* (blue), and of only 1.17 ± 0.25 fold for γ8 S72\* (red). Statistical significance was analyzed using a two-tailed unpaired Welch's $t$ test; \*$p < 0.05$. **h** Plot of all the analyzed spines fluorescent intensities as a function of the intensity in a corresponding neighboring equivalent extraspine area in the dendrite for γ2 S44\* (blue) and γ8 S72\* (red) expressing neurons. The Pearson's correlation coefficients are γ2 S44\*: blue, $r = 0.57$, $p < 0.0001$, 872 spines; and, γ8 S72\*: red, $r = 0.81$, $p < 0.0001$, 521 spines. **i** Plot of the average ratio per neuron of spine to extraspine intensities as a function of the sum of spine and extraspine intensities for γ2 S44\* (blue, $r = 0.13$, $p = 0.521$) and γ8 S72\*: and γ8 S72\* (red, $r = 0.-26$, $p = 0.189$) expressing neurons. (**g–i**) Data relative to γ2 S44\* and γ8 S72\* pulled from 28 cells each from three independent biological replicates. All representative images are representative of three independent preparations. Source data are provided as a Source Data file.

both TARPs in and outside of synapses randomly distributed localization clusters with a size of ~20 nm, which can be attributed to multiple localized Cy5 dye molecules. Only γ2 S44\* in synaptic areas showed strong deviation from the simulations with a maximum at ~100 nm indicating cluster formation (Supplementary Fig. 6). Individual cluster analysis for each ROI in and outside of synapses confirmed the existence of γ2 S44\* clusters in synapses and the absence of extrasynaptic γ2 S44\* and γ8 S72\* clusters (Fig. 5d). In addition, the synaptic ROIs exhibited a higher localization density for γ2 S44\* clusters with an average size of ~80 nm (Fig. 5c, d and Supplementary Fig. 7).

## Discussion

The ability to label target proteins with small ligands, and at sterically hard-to-access epitopes, represents an important challenge in biology, in particular for live-cell and super-resolution imaging studies in neurons. TARPs represent an interesting case study as their limited extracellular loops and close association with AMPARs has prevented the development of adequate ligands, in particular for the study of TARPs organization and trafficking at the cell surface of living neurons. Our motivation to search for alternative labeling strategies was further reinforced by our initial finding that functional antibodies to the extracellular domains of γ2 and γ8 were unable to recognize native TARPs in neurons, likely due to epitope masking. We thus engineered the technology to incorporate ncAAs in these proteins at given edited sites by GCE to label them directly with fluorophores by click chemistry. Due to the potential of GCE for protein tagging, and emerging interest in using such strategy for protein labeling via click chemistry, in particular in the context of neuroscience[51], we worked with commercially available reagents for reproducibility and broader reach and developed a pipeline to label neurons in two different model systems, including cultured brain slices that preserve the physiological network environment.

We demonstrate that γ2 and γ8 can be directly labeled with this approach both in dissociated primary cultures and organotypic slices of rodent hippocampal neurons. Labeled proteins can then be imaged by a panoply of different approaches, including widefield, confocal, or *d*STORM super-resolution imaging due to the vast combination possibilities with different tetrazine-dyes. As some tetrazine-dyes tend to bind non-specifically to intracellular compartments, previous work show that it is important to carefully select suitable tetrazine-dyes and establish proper control experiments for each specific application[33].

Our data reveal that γ2 and γ8 display profoundly different distributions on the neuronal surface, γ2 being much more concentrated and clustered at synapses than γ8. In addition, the ability to label the masked epitopes in close proximity to the associated AMPAR subunits allowed us to develop FRET pairs between γ2 or γ8 and GluA subunits. Further development of the

FRET pairs with smaller tags on AMPARs, such as α-bungarotoxin binding site-tag[52] and development of CRISPR/Cas9 technology for the site-specific incorporation of a single ncAA will be of value for the study of AMPAR and TARP dynamic interaction in live neurons, in particular, in the context of synaptic plasticity. Of note, we recently used GCE in combination with self-labeling enzymes to study the association/dissociation of heterodimers at the cell surface[52]. We, therefore, envision multiple applications of FRET-based sensors to study the dynamics of the AMPAR-TARP interactions in the future.

In the attempt to label surface TARPs, we first developed antibodies against the extracellular domains of γ2 and γ8. While live labeling with our antibodies against γ2 Ex2 or γ8 Ex1 was able to specifically detect the respective recombinant protein in cell lines and dissociated neurons, this approach failed to detect endogenous TARPs in dissociated hippocampal neurons or recombinant γ2 genetically tethered to GluA2. This indicates that the extracellular loops of γ2 and γ8 are masked when associated or in contact with AMPAR, which is compatible with the published cryo-EM structures of TARP/GluA subunit complexes[21–23]. Importantly, it further indicates that, at endogenous levels, most if not all γ2 and γ8 are associated with AMPAR on the surface of hippocampal neurons, as γ2 in particular only associates to AMPAR subunits[27]. This had remained an important open question in the field. This result does not preclude the potential existence of intracellular AMPAR-free TARPs, particularly along the biosynthetic pathway[53], a point not addressed by our study. Small tags like hemagglutinin (HA, 9 aa)[27] or biotin-acceptor peptide (bAP, 15 aa) were previously successfully inserted in the Ex1 loop of γ2, while the incorporation of bigger proteins such as mCherry in the Ex1 led to intracellular retention of γ2 possibly due to protein misfolding[28]. Surface labeling of γ2-bAP with streptavidin[28] or γ2-HA with specific antibodies[12,54] in the extracellular loops had previously been achieved in neurons, but most likely only revealed overexpressed TARP not associated with AMPARs.

In contrast, GCE combined with click chemistry labeling allowed the site-specific incorporation of ncAAs that can be functionalized and labeled with small tetrazine-dyes with a size of ~1 nm[33]. Both patch-clamp and FRET experiments demonstrate that GCE-labeled TARPs are fully functional and can interact normally with GluA subunits. Indeed, electrophysiological recordings show that the incorporation of ncAAs into the Ex1 of γ2 and γ8 did not compromise TARP-specific AMPAR gating modulation, while FRET experiments indicate close association between ncAA-tagged TARPs and GluA subunits. A parallel approach using cysteine tagging of AMPAR and ncAA-tagging of γ2 TARPs enabled luminescence resonance energy transfer and single-molecule FRET live cell measurements of the distance between GluA2 and γ2 in HEK293T cells and the study of its regulation[20]. Worth mentioning, while we did not observe a

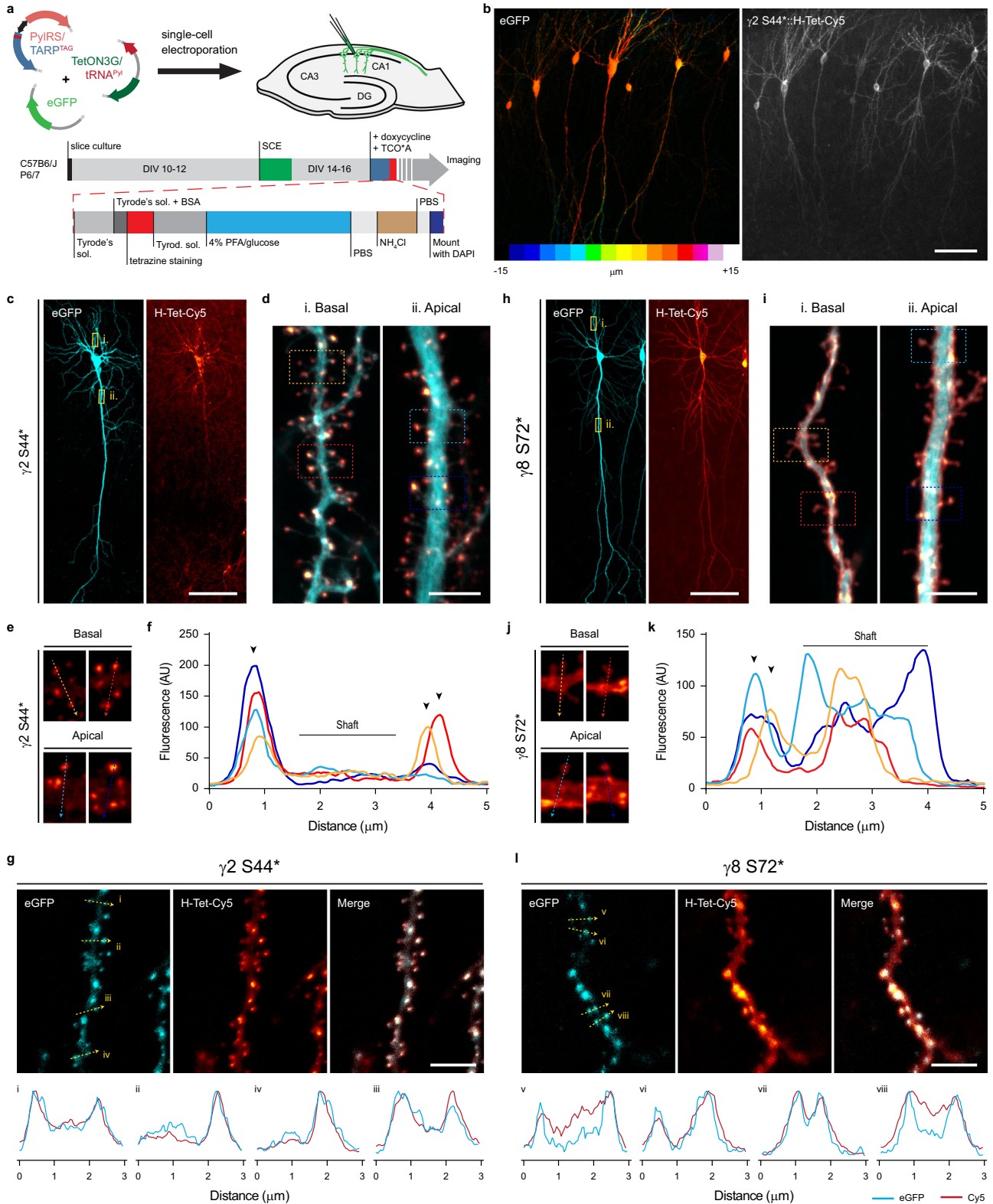

difference in terms of labeling efficacy among tested Pyr-Tet-dyes and H-Tet-dyes in both cell lines and dissociated neurons, we did observe that H-Tet-Cy5 outperformed Pyr-Tet-ATTO643 in OHSC. This observation could be explained by the faster reaction of H-Tet with TCO*A as well as the lower sterical demand compared to Pyr-Tet[55,56]. We also observed some decrease in H-Tet-Cy5 fluorescence intensity with depth in organotypic slices, usually accompanied by a decrease in eGFP fluorescence

intensity, suggesting inefficient excitation due to scattering issues rather than inefficient tetrazine-dye labeling.

Previous EM studies have suggested γ2 plasma membrane distribution to be almost exclusively synaptic, with γ8 being more equally distributed between extrasynaptic and synaptic sites[18,19]. It is interesting to note however that TEM could only detect γ2 and γ8 peri-synaptically[19], likely due to epitope masking. The limits inherent to EM (sensitivity, antigen accessibility) thus make

**Fig. 4 Bioorthogonal labeling of γ2 S44\* and γ8 S72\* in organotypic hippocampal slice cultures report a distinct surface distribution of TARPs.**
**a** Depiction of the workflow used for expression of ncAA-tagged TARPs in CA1 pyramidal cells in OHSC using single-cell electroporation (SCE), and live staining with tetrazine-dyes. **b** Example confocal image of fixed CA1 neurons co-expressing eGFP and γ2 S44\* in OHSC. Images are projections of a z-stack taken by 1 μm increments, eGFP signal is color-coded with respect to sample depth. **c**, **e** Representative confocal images of CA1 neurons co-expressing eGFP, and **c** γ2 S44\*- or **h** γ8 S72\* live stained with 1 μM H-Tet-Cy5. **d**, **f** Magnified views of segments of the basal and apical dendrites from **d** γ2 S44\*- and **f** γ8 S72\*-overexpressing CA1 neurons highlighted in the corresponding overview images (yellow boxes). **g**, **i** Close up of representative spines from **g** γ2 S44\*- and **i** γ8 S72\*-overexpressing CA1 neurons highlighted (dashed squares) in the overview images (**d**) and (**f**), respectively. **h**, **j** Line scan measurements of Cy5 signal across spines in **g** and **i** respectively. **k**, **l** Confocal images of segments of basal dendrites from CA1 neurons co-expressing either **k** γ2 S44\* or **l** γ8 S72, and the PSD-95 marker, XPH20::eGFP. Bottom insets: line scans of the GFP and Cy5 signal for the 3 μm segments indicated in the above images. Scale bar: (**b**, **c**, **h**) 100 μm and (**d**, **g**, **i**, **l**) 5 μm. (**b**–**e**, **h**–**j**) example images are representative of three or four independent preparations, and (**g**, **l**) from two independent preparations.

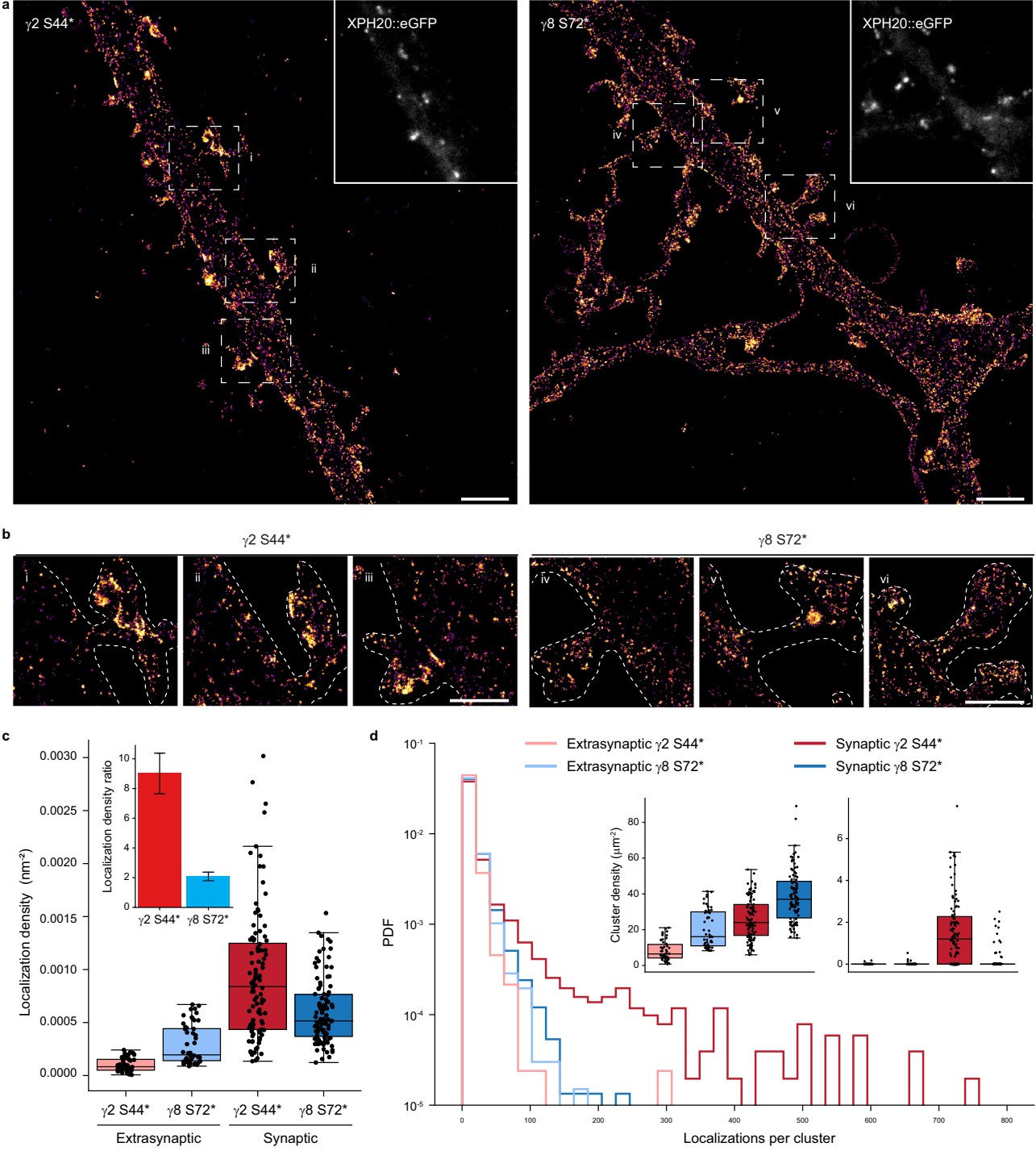

**Fig. 5 dSTORM imaging and analysis reveal nanoscale organization of bioorthogonal labeled γ2 S44* and γ8 S72* in dissociated neurons.**
**a** Representative dSTORM image of Pyr-Tet-AF647 (0.5 µM) live labeled neurons expressing γ2 S44* or γ8 S72* co-expressed with XPH20::eGFP. Scale bar: 2 µm. **b** Magnified views of spine and dendrite of the respective overview images in **a** (dashed rectangles). Scale bar: 1 µm. **c** Boxplots displaying higher synaptic localization densities for γ2 S44* (($0.93 \pm 0.06$)*10E-3 $nm^{-2}$, $n = 104$, dark red) compared to γ8 S72* (($0.60 \pm 0.03$)*10E-3 $nm^{-2}$, $n = 102$, dark blue). γ8 S72* showed higher extrasynaptic localization densities (($0.29 \pm 0.03$)*10E-3 $nm^{-2}$, $n = 52$, light blue) in comparison to γ2 S44* (($0.10 \pm 0.01$) *10E-3 $nm^{-2}$, $n = 50$, light red). Inset: ratio of synaptic to extrasynaptic mean localization densities indicate a spine enrichment of $9.0 \pm 1.4$ folds for γ2 S44* (red) and of $2.1 \pm 0.3$ for γ8 S72* (blue). **d** Histograms showing localizations number per cluster for synaptic (dark), extrasynaptic (light) γ2 S44* (red), and γ8 S72* (blue), displayed as probability density function (PDF) ($n = 2039, 3243, 2486, 3644$ cluster from 50, 52, 104, 102 ROIs of five preparations for extrasynaptic γ2, extrasynaptic γ8, synaptic γ2, synaptic γ8, respectively). Insets display boxplots of ROI cluster densities for clusters with less (left inset) and more than 100 clustered localizations (right inset). Only synaptic γ2 S44* shows clusters with >100 localizations ($1.93 \pm 0.19\ \mu m^{-2}$) compared to nearly no clusters for synaptic γ8 S72* ($0.21 \pm 0.06\ \mu m^{-2}$) and extrasynaptic γ2 S44* ($0.01 \pm 0.01\ \mu m^{-2}$) or γ8 S72* ($0.03 \pm 0.01\ \mu m^{-2}$). For a selection of clusters with <100 localizations, γ8 S72* presents larger densities in synaptic ($38 \pm 1\ \mu m^{-2}$) as well as extrasynaptic areas ($20 \pm 2\ \mu m^{-2}$) in comparison to γ2 S44* clusters (synaptic: $26 \pm 1\ \mu m^{-2}$, extrasynaptic: $8 \pm 0.8\ \mu m^{-2}$). Boxplots show lower to upper quartile and median values of the data with whiskers extending 1.5 × interquartile range. All data represent mean ± SEM. Source data are provided as a Source Data file.

the development of TARP labeling tools applicable in light microscopy even more relevant. In addition, functional studies have indicated that γ2 promotes synaptic targeting of AMPARs[7,11,12] whereas γ8 controls extrasynaptic surface pool and synaptic delivery of AMPARs[9,18]. Furthermore, at Schaffer collateral/commissural (SCC) synapses in the adult mouse hippocampal CA1, synaptic inclusion of γ2 potently increases AMPAR expression, and transforms low-density synapses into high-density ones, whereas γ8 is essential for low-density or basal expression of AMPARs at non-perforated synapses[35], which is fully compatible with our observations. Therefore, these TARPs are critically involved in AMPAR density control at SCC synapses. However, specific imaging of γ2 and γ8 distribution in live neurons was lacking due to the absence of adequate tools. Our data indicate that both in dissociated hippocampal and organotypic CA1 pyramidal neurons, γ2 S44* shows a strong accumulation and forms clusters with a size of ~80 nm at spines compared to lower appearance and more homogeneous distribution at the dendritic shaft. In contrast, γ8 S72* shows a more homogenous distribution between spines and dendritic shaft without any indication of cluster formation.

As mentioned, a limitation in our study is the fact that we had to use an overexpression approach. Recent advances in genome editing tools, such as CRISPR/Cas9, will likely make it possible in the future to deliver site-specific incorporation of ncAAs into endogenous proteins in post-mitotic cells, such as neurons. The combination of this approach with future whole-genome recoding in which all the endogenous Amber codons are replaced by Ochre codons[57] would be particularly valuable. Another alternative that might be more reachable in the near future is the use of orthogonal ribosomes[58,59] combined with quadruplet codons[60], eliminating the possibility of tRNA-induced suppression of endogenous Amber codons as well as improving the incorporation of ncAA. In a complementary work to ours in preprint, Arsíc and colleagues[61] showed the potential of using bioorthogonal labeling to tag intracellular proteins in live neurons using a similar approach, further expanding the versatility and high potential of GCE in the context of neuroscience. Using a second tag (FLAG-tag or GFP) carrying an Amber codon mutation, these authors could incorporate ncAAs into endogenous proteins using CRISPR/Cas9 strategy. However, this strategy relies on the use of 'conventional' tags to deliver the Amber codon at the C-terminus of the target protein and lacks the versatility to site-specific incorporation of ncAAs.

In conclusion, the robustness and versatility of the approach shown here, and the panoply of cell-permeable and impermeable tetrazine-dyes[33] opens a spectrum of possibilities that will be fascinating to explore, including for multicolor imaging of the nanoscale organization, interactions, and trafficking of

intracellular and/or extracellular proteins in living neurons. The minimal perturbation of the target protein by insertion of a single ncAA and small size of tetrazine-dyes enables stoichiometric labeling even of sterically shielded protein sites. The method will thus be particularly valuable for quantitative super-resolution microscopy as it provides a sterically minimally demanding labeling and in principle, a perfectly controlled stoichiometric labeling as TCO*-tetrazine labeling exhibits a ratio of 1, and each tetrazine is labeled with a single dye. Additionally, due to the wide range of tetrazine-dyes available nowadays, this tool can be easily combined with other smaller tags, like HA- or bAP-tag. A limitation however remains in the capacity to demonstrate a saturation of the labeling. Multicolor GCE has been achieved[62,63], but is still challenging as to achieve dual-color labeling with two different ncAAs requires not only two orthogonal click-reactions but also need two mutually orthogonal tRNA/RS pairs which can specifically incorporate two distinctly clickable ncAAs. While multicolor labeling using two mutually orthogonal tRNA/RS pairs capable of specifically incorporate two distinctly clickable ncAAs is difficult with the current technology, GCE can easily be combined with other labeling strategies, including relatively small tags.

Altogether, bioorthogonal labeling of TARPs in living neurons constitutes an important achievement in protein tagging in the field of neuroscience, as it not only introduces a robust and fast labeling strategy with minimal to no-perturbation but also allows the labeling of hard-to-access proteins that to date have been highly affected by the bulky size of previous labeling strategies[52]. This altogether opens the possibility to tackle new sets of biological questions.

## Methods

**Reagents**. Trans-Cyclooct-2-en-L-Lysine (TCO*A; #SC-8008) was purchased from SiChem (Bremen, Germany). Pyrimidyl-Tetrazine-Alexa Fluor 647 (Pyr-Tet-AF647; #CLK-102), Pyr-Tet-ATTO-643 (Pyr-Tet-ATTO643; #CLK-101), H-Tet-Cy3 (#CLK-014-05), and H-Tet-Cy5 (#CLK-015-05) were purchased from Jena Bioscience (Jena, Germany). SNAP-Surface® Alexa Fluor® 488 (BG-AF488; #S9129S) was purchased from New England Biolabs. 2,3-Dioxo-6-nitro-1,2,3,4-tetrahydrobenzo[f]quinoxaline-7-sulfonamide disodium salt (NBQX; #1044) and Kainate (KA; #0222) were purchased from Tocris. L-Glutamic acid monosodium salt (Glu; #G1626) and doxycycline (#D1822) were purchased from Sigma.

**Plasmid constructs**. Plasmid amplification was performed via transformation in *E. coli* DH5α (Thermo Fisher Scientific, #EC0111) or *E. cloni*® 10G (Lucigen, #60107) in the case of pTRE3G plasmids, and DNA isolation via MAXI-prep ZymoPURE II Plasmid kits (Zymo Research).

eGFP, mCherry or mEos2 were cloned into the coding sequence of γ2 (between residues 304 and 305) and γ8 (between residues 401 and 402) by introducing AgeI/NheI sites to the respective position. The respective Amber stop mutants (Supplementary Fig. 1c) were generated by introducing a TAG codon through PCR-based site-directed mutagenesis in pcDNA3 vector. For γ8, the endogenous TAG stop codon of WT γ8 was replaced by a TAA stop codon. The plasmid for the

expression of the tRNA/aminoacyl transferase pair (pCMV tRNA$^{Pyl}$/NESPylRS$^{AF}$, herein termed PylRS/tRNA$^{Pyl}$) was kindly provided by Edward Lemke[64].

The NESPylRS$^{AF}$ was inserted into a bidirectional doxycycline-inducible expression vector pTRE3G-BI (Takara Bio, #631332), herein termed pTRE3G-BI PylRS, using the restriction sites BamHI/BglII into the BamHI restriction site of the multiple cloning site of pTRE3G-BI after PCR amplification using the oligonocleotides: PylRS_F, 5′-CTTGGATCCGCCACCATGGATAAAAAACC-3′ and PylRS_R, 5′-TAGAAGCTTTTACAGGTTAGTAGAAATACCATTGTAATAG-3′.

To reduce TARPs expression toxicity in neurons, and reduce the number of plasmids to transfect, WT TARPs and ncAA-tagged TARPs were subcloned into the plasmid pTRE3G-BI PylRS using the restriction sites KpnI/XbaI.

The U6 promoter and tRNA$^{Pyl}$ were inserted into the pEF1α-Tet3G (Takara Bio, #631336; Tet3G/tRNA) using the restriction site BsrGI after PCR amplification using the oligonucleotides: U6/tRNA_F, 5′-GCATGTACATTTCCCCGAAAAATGG-3′ and U6/tRNA_R, 5′-GGTCATATTGGACATGAGCC-3′ (primer located upstream the U6 promoter on the pCMV tRNAPyl/NESPylRSAF), and co-expressed with the pTRE3G-BI constructs.

γ2::eGFP and tethered GluA2 (flop isoform)::γ2::eGFP[65] were subcloned into the doxycycline-inducible expression vector pTRE3G-BI (Takara Bio, #631332) using the restriction sites XbaI/BamHI and EagI/BamHI, respectively.

MfeI/NheI restriction sites were introduced after the signal peptide of GluA1 to insert the SNAP-tag® at N-terminus of GluA1 (SNAP::GluA1) flip variant coding sequence in pRK5 vector.

The plasmid for the GluA1 Tn5 ME SEP + 396 aa was kindly provided by Andrew Plested. AgeI/NheI restriction sites were introduced between the Tn5 ME sequences and SEP was replaced by SNAP-tag® (GluA1::SNAP396). The tethered GluA1::SNAP396::γ2 S61* and GluA1::SNAP396::γ8 K102* were performed as described for the tethered WT GluA1::γ2 in Morimoto-Tomita et al.[37].

The plasmid for the expression of the tRNA/aminoacyl transferase pair (pNEU-hMbPylRS-4xU6M15, herein termed PylRS/4xtRNA$^{Pyl}$) was a gift from Irene Coin (Addgene, #105830)[40].

The plasmid for the expression of the Xph20 eGFP CCR5TC (XPH20::eGFP) was a gift from Matthieu Sainlos[47,48].

The plasmid for the expression of GFP39TAG (herein termed clickable-GFP) was kindly provided by Edward Lemke[39].

**Heterologous cell culture.** HEK293T cells (ECACC, #12022001) were cultured at 37 °C under 5% $CO_2$ in DMEM supplemented with 10% FBS, 1% L-glutamine, and 1% penicillin/streptomycin. COS-7 cells (ECACC, #87021302) were cultured at 37 °C under 5% $CO_2$ in DMEM supplemented with 10% FBS, 1% L-glutamine, and 1% penicillin/streptomycin.

**Animals.** All experiments were performed in accordance with the European guidelines for the care and use of laboratory animals, and the guidelines issued by the University of Bordeaux animal experimental committee (CE50; Animal facilities authorizations A3306940 and A33063941).

Tissue for dissociated hippocampal cultures was harvested from embryos of an unascertained mixture of sexes prevenient from gestant Sprague-Dawley rat females at the age of 9–12 weeks old purchased weekly from Janvier Labs, Saint-Berthevin, France. Tissue for OHSC was harvested from WT C57Bl6/J mice of both sexes at postnatal day 5–7 raised at PIV-EOPS facility of the IINS. Animals were housed at PIV-EOPS facility of the IINS under a 12 h light/dark cycle at normal room temperature (22 °C) and humidity between 40 and 70% (typically 60%) with unrestricted access to food and water.

**Primary dissociated hippocampal neurons.** Dissociated hippocampal neurons from embryonic day 18 (E18) Sprague-Dawley rats embryos of both sexes were prepared as previously described[66]. Briefly, dissociated neurons were plated at a density of 250,000 cells per 60 mm dish on 0.1 mg mL$^{-1}$ PLL pre-coated 1.5H, ∅ 18 mm coverslips (Marienfeld Superior, #0117580). Neurons cultures were maintained in Neurobasal™ Plus Medium (Thermo Fisher Scientific) supplemented with 0.5 mM GlutaMAX (Thermo Fisher Scientific) and 1X B-27™ Plus Supplement (Thermo Fisher Scientific). D-2 μM Cytosine β-D-arabinofuranoside (Sigma Aldrich) was added after 72 h. At DIV3/4, cells were transfected with the respective cDNAs using Lipofectamine 2000 (Thermo Fisher Scientific, #11668019). Cultures were kept at 37 °C under 5% CO2 up to 18 days.

Astrocytes feeder layers were prepared from the similar embryos, plated between 20,000 and 40,000 cells per 60 mm dish and cultured in Minimum Essential Medium (Thermo Fisher Scientific) containing 4.5 g L$^{-1}$ glucose, 2 mM GlutaMAX, and 10% heat-inactivated horse serum for 14 days.

**Organotypic hippocampal slice cultures.** OHSC from animals at postnatal day 5–7 from wild type mice of both sexes (C57Bl6/J strain) were prepared as previously described[67]. Briefly, animals were quickly decapitated and hippocampi were dissected out and placed in ice-cold carbonated dissection buffer (in mM): 230 sucrose, 4 KCl, 5 $MgCl_2$, 1 $CaCl_2$, 26 $NaHCO_3$, 10 D-glucose, and phenol red. Coronal slices (300 μm) were cut using a tissue chopper (McIlwain), collected and

positioned on interface-style Millicell® culture inserts (Millipore) in six-well culture plates containing 1 mL of sterile serum-containing MEM medium (in mM): 30 HEPES, 5 $NaHCO_3$, 0.511 sodium L-ascorbate, 13 D-glucose, 1 $CaCl_2$, 2 $MgSO_4$, 5 L-glutamine, and 0.033% (v/v) insulin, pH 7.3, osmolality adjusted to 317–320 mOsm, plus 20% (v/v) heat-inactivated horse serum. Brain slices were incubated at 35 °C under 5% $CO_2$ and the culture medium was changed from the bottom of each well every 2–3 days. After 11–13 days in culture, slices were transferred to an artificial cerebrospinal fluid (ACSF) containing (in mM): 130 NaCl, 2.5 KCl, 2.2 $CaCl_2$, 1.5 $MgCl_2$, 10 D-glucose, and 10 HEPES, pH 7.35, osmolality adjusted to 300 mOsm. CA1 pyramidal cells were then processed for single-cell electroporation (SCE)[68] using glass micropipettes containing K-gluconate-based intracellular solution (in mM): 135 K-gluconate, 4 NaCl, 2 $MgCl_2$, 2 HEPES, 2 $Na_2ATP$, 0.3 NaGTP, 0.06 EGTA, 0.01 $CaCl_2$ (pH 7.2–7.3 with KOH, osmolality adjusted to 290 mOsm) with plasmids encoding Tet3G/tRNA$^{Pyl}$ and pTRE3G-BI PylRS/γ2 S44* or pTRE3G-BI PylRS/γ8 S72* in equal proportions (26 ng μl$^{-1}$) along with eGFP (13 ng μl$^{-1}$) or XPH20 eGFP (13 ng μl$^{-1}$). Patch pipettes were pulled from 1 mm borosilicate capillaries (Harvard Apparatus) with a vertical puller (Narishige, #PC-100). SCE was performed by applying four square pulses of negative voltage (−2.5 V, 25 ms pulse width) at 1 Hz. After SCE, slices were placed back in the incubator for 4–5 days before labeling.

**Electrophysiology.** cDNAs for GluA1 (250 ng), PylRS/tRNA$^{RS}$ (375 ng), and WT/ncAA-tagged γ2/γ8 eGFP or soluble eGFP (375 ng) were co-transfected into HEK293T cells (90,000–100,000 cells per cm$^{-2}$ in 12-well plate) using jetPRIME® (Polyplus-transfection, #114-01). 250 μM TCO*A and 40 μM NBQX were added to the cells at the time of the transfection. Cells were trypsinized 1 day after transfection and seeded on PLL-coated coverslips. Cells were transferred to the recording chamber, and brightly fluorescent isolated cells were selected. Whole-cell patch-clamp recordings were performed at room temperature in HEPES-buffered Tyrode's solution (HBSS) containing (in mM): 138 NaCl, 2 KCl, 2 $MgCl_2$, 2 $CaCl_2$, 10 D-glucose, and 10 HEPES, pH 7.4, osmolality adjusted to 317–320 mOsm. Patch pipettes were filled with an internal solution containing (in mM): 120 $CsCH_3SO_3$, 2 NaCl, 2 $MgCl_2$, 10 EGTA, 100 HEPES, and 4 $Na_2ATP$, pH 7.4, osmolality 312 mOsm. Pipette resistances for these experiments were typically 3–5 MΩ and cells with a series resistance higher than 15 MΩ were discarded. Glu (10 mM) or KA (0.1 mM) were dissolved in HEPES-buffered solution and applied using a theta pipette driven by a piezoelectric controller (Burleigh, #PZ-150M). Membrane potential was held at −60 mV. Currents were collected using an EPC10 amplifier (HEKA) and filtered at 2.9 kHz and recorded at a sampling frequency of 20 kHz.

**TARPs immunostaining.** cDNAs for Stg mEos2 or γ8 mEos2 (500 ng) were transfected into COS-7 cells (14,000–17,000 cells per cm$^2$ in 12-well plate) for 24 h using X-tremeGENE HP DNA (Roche, #06366236001). Cells were incubated for 7 min at 37 °C with either 4 μg mL$^{-1}$ rabbit anti-γ2 Ex2 or 1:50 serum rabbit anti-γ8 Ex1 antibodies before fixation. Dissociated hippocampal neurons were co-transfected either with pTRE3G-BI γ2::eGFP or tethered pTRE3G-BI GluA2::γ2::eGFP, and Tet3G/tRNA$^{Pyl}$ in equal proportions (125 ng). Transfected neurons were treated with 200 ng mL$^{-1}$ doxycycline 18 h before use. Neurons were incubated for 7 min at 37 °C with 10 μg mL$^{-1}$ mouse anti-GluA (Synaptic Systems, #182411) and anti-γ2 Ex2 or anti-γ8 Ex1 antibodies before 4% PFA/sucrose fixation. Reactive aldehydes groups were blocked for 10 min with 50 mM $NH_4Cl$. Alternatively, neurons were live incubated with the anti-GluA antibody, and after fixation neurons were permeabilized with 0.2% Triton-X100 for 5 min and incubated with 0.4 μg mL$^{-1}$ rabbit anti-γ8 antibody (Frontiers Institute, #TARPg8-Rb-Af1000) diluted in 3% BSA in PBS. Cells were incubated with the respective secondary antibodies anti-mouse AF568 and anti-rabbit AF647 (Thermo Fisher Scientific) diluted at 1:1000 in 3% BSA in PBS. Imaging was performed on an up-right widefield fluorescence microscope (Leica Microsystems, Leica DM5000 B) microscope controlled by Metamorph software (Molecular Devices). Fluorescence excitation of eGFP, AF568 and AF647 was done by a LED SOLA Light (Lumencor). Images were acquired using an oil-immersion objective (Leica, HCX PL APO 40x/NA 1.25 OIL) and appropriate filter set. Fluorescent emission was collected using a sCMOS camera (Hamamatsu Photonics, ORCA-Flash4.0 V2).

**Bioorthogonal labeling in HEK293T cells.** HEK293T cells plated at a density of 80,000–90,000 cells–cm$^{-2}$ on a pre-coated PDL 4-well Nunc™ Lab-Tek™ II chamber (Thermo Fisher Scientific, #155382PK) were co-transfected with PylRS/4xtRNA$^{Pyl}$ (500 ng) and respective tagged TARPs (500 ng) using jetPRIME® transfection reagent for 24 h in the presence or absence of 250 μM TCO*A. Cells were washed once with cell media to remove excessive TCO*A prior to labeling with 1.5 μM Pyr-Tet-ATTO643 or H-Tet-Cy5 diluted in TCO*A-free medium for 30 min on ice. Subsequently, cells were rinsed three times with ice-cold HBSS and immediately live imaged or fixed for 15 min at RT with 4% FA in PBS followed by three washing steps with HBSS before imaging. Confocal imaging of living or fixed cells was performed using a LSM700 setup (Zeiss) equipped with an oil-immersion objective (Zeiss, Plan-Apochromat 63x/NA 1.4 OIL). eGFP and Pyr-Tet-ATTO643/H-Tet-Cy5 were excited using a 488 nm or 641 nm solid-state laser and respective filter settings. Images were processed in ImageJ (FIJI) adjusting brightness and contrast to identical values for comparison of experiments.

**Bioorthogonal labeling in dissociated hippocampal neurons.** Dissociated hippocampal neurons were co-transfected with Tet3G/tRNA$^{Pyl}$ (104 ng), pTRE3G-BI PylRS/TARPs (γ2, γ8, γ2 S44* or γ8 S72*; 104 ng), along with eGFP or XPH20 eGFP (42 ng) at DIV 3–4 using lipofectamine 2000. At DIV16-18, 250 μM TCO*A and 100 ng mL$^{-1}$ doxycycline were added to the cell media for a period of ~20 h. Alternatively, neurons were co-transfected with Tet3G/tRNA$^{Pyl}$ (104 ng), pTRE3G-BI PylRS (104 ng), and clickable-GFP (42 ng) at DIV 3-4. Five days prior to H-Tet-Cy5 labeling, 100 ng mL$^{-1}$ doxycycline and 250 μM TCO*A were added to the cell media. An extra 50 ng mL$^{-1}$ doxycycline and 125 μM TCO*A were added 24 h before labeling upon replacing half the media by fresh one. Cells were rinsed three times with warm Tyrode's solution containing (in mM): 100 NaCl, 5 KCl, 5 MgCl$_2$, 2 CaCl$_2$, 15 D-glucose, and 10 HEPES, pH 7.4, osmolarity adjusted to 243–247 mOsm followed by 3 min incubation in Tyrode's solution containing 1% BSA. Cells were then incubated with 0.5 μM tetrazine-dye for 7 min at 37 °C and rinsed four times with Tyrode's solution.

Live-cell imaging was performed in Tyrode's solution at 37 °C using an incubator box with an air heater system (Life Imaging Services) installed on an inverted Leica DMI6000 B (Leica Microsystem) spinning disk microscope controlled by Metamorph software (Molecular Devices). Z-stacks of whole neurons were acquired using an oil-immersion objective (Leica, HCX PL APO ×40/NA 1.25 OIL) and appropriate filter set. Fluorescent emission was collected using a sCMOS camera (Hamamatsu, ORCA-Flash4.0 V2).

Alternatively, cells were fixed for 10 min using 4% PFA/glucose. Reactive aldehydes groups were blocked for 10 min with 50 mM NH$_4$Cl. Images of fixed neurons were acquired with a Leica TCS SP8 confocal microscope controlled by Leica Application Suite X (LAS X) software and equipped with hybrid detectors. eGFP and Pyr-Tet-ATTO643 were excited at 488 nm and 638 nm, respectively. For quantification of γ2 S44* and γ8 S72* surface distribution in dissociated hippocampal neurons, Z-stacks of whole dendrite segments were acquired using an oil-immersion objective (Leica, HC PL APO CS2 63x/NA1.40 OIL) and a pinhole opened to one time the Airy disk.

**Bioorthogonal labeling in OHSC.** Single electroporated neurons from OHSC co-expressing pTRE3G-BI PylRS/γ2 S44* or pTRE3G-BI PylRS/γ8 S72*, Tet3G/tRNA$^{Pyl}$ and eGFP were treated with 250 μM TCO*A and 100 ng mL$^{-1}$ doxycycline for ~22 h before labeling. Slices were washed three times 5 min with warm ACSF followed by 5 min in ACSF containing 1% BSA. Subsequently, slices were incubated for 10 min at 35 °C with 1 μM H-Tet-Cy5 diluted in ACSF containing 1% BSA and washed four times 5 min with ACSF. Slices were fixed for 2 h at RT with 4% PFA/sucrose, washed with PBS. Reactive aldehydes groups were blocked for 20 min in 200 mM NH$_4$Cl. Slices were mounted in Fluoromount-G Mounting Medium (Thermo Fisher Scientific, #00-4958-02) and left to cure for 48 h at RT before imaging.

Images of fixed neurons were acquired with a Leica TCS SP8 confocal microscope controlled by Leica Application Suite X (LAS X) software and equipped with hybrid detectors. eGFP and Pyr-Tet-ATTO643 were excited at 488 nm and 638 nm, respectively. Z-stacks of whole neuron were acquired using an oil-immersion objective (Leica, ×20/NA 0.70 IMM) and a pinhole opened to two times the Airy disk. For quantification of γ2 S44* and γ8 S72* surface distribution, Z-stacks of segments basal and apical dendrite were acquired using an oil-immersion objective (Leica, HC PL APO CS2 ×63/NA 1.40 OIL) and a pinhole opened to one time the Airy disk.

**Quantification of γ8 overexpression.** To determine γ8 overexpression levels upon transfection with γ8 S72*, dissociated hippocampal neurons were transfected with Tet3G/tRNAPyl (104 ng), pTRE3G-BI PylRS/γ8 S72* (104 ng), along with XPH20 eGFP (42 ng); TCO*A and doxycycline was added to the media ~20 h prior to tretrazine labeling. Surface γ8 S72* was labeled with 0.5 μM H-Tet-Cy5 as above described. Upon fixation, cells were permeabilized and incubated with the anti-γ8 antibody (Frontiers Institute, #TARPg8-Rb-Af1000) (see TARPs immunostaining section). Cells were imaged using an inverted Leica DMI6000 B (Leica Microsystem) spinning disk microscope controlled by Metamorph software (Molecular Devices). Z-stacks of whole neurons were acquired using an oil-immersion objective (Leica, HCX PL APO 40x/NA 1.25 OIL) and appropriate filter set. Fluorescent emission was collected using a sCMOS camera (Hamamatsu, ORCA-Flash4.0 V2).

All images were analyzed using ImageJ (FIJI) software. Images of non-transfected and γ8 S72*-positive neurons were maximum intensity Z-projected. Masks of regions of interest (dendritic tree) generated based on AF568 (anti-γ8) images upon a median filter (radius = 1) were applied. Relative fluorescence intensity was calculated based on the average fluorescence intensity of non-trasfected cells.

**γ2 S44* and γ8 S72* surface distribution in neurons.** All images were analyzed using ImageJ (FIJI) software. Confocal images of dissociated neurons co-expressing eGFP, Tet3G/tRNA$^{Pyl}$ and pTRE3G-BI PylRS/TARPs (γ2, γ8, γ2 S44*, or γ8 S72*) were maximum intensity Z-projected. For tetrazine specificity, 3 pixel-width line scans across spines and dendritic shaft and cell-free areas were performed based on eGFP fluorescence. For surface distribution, masks of regions of interest (spine and adjacent dendritic draft area) generated based on thresholded eGFP images upon a Gaussian blur filter (radius = 1) were applied. Spine enrichment was calculated as the mean spine fluorescence intensity over the neighbor dendritic area mean fluorescence.

For surface distribution of γ2 S44*, γ8 S72*, and XPH20::eGFP in OHSC, confocal images of dendritic segments were integrated intensity Z-projected. Upon a Median filter (radius = 1) was applied, 3 pixel-width line scans across spines that were perpendicular to the dendritic shaft were performed based on eGFP fluorescence.

Confocal images of dissociated neurons co-expressing clickable-GFP, Tet3G/tRNA$^{Pyl}$ and pTRE3G-BI PylRS labeled with H-Tet-Cy5 were maximum intensity Z-projected. For the purpose of accessing possible off-target surface labeling, 5 pixel-width line scans across random regions in the field-of-view were performed based on clickable-GFP fluorescence.

**dSTORM imaging.** The TARP constructs γ2 S44* or γ8 S72*-positive neurons at DIV17-18 co-transfected with XPH20::eGFP were live stained with 0.5 μM Pyr-Tet-AF647 and fixed with 4% FA and 0.25% GA in PBS for 15 min.

The dSTORM images were acquired using an inverted wide-field fluorescence microscope (Olympus, IX-71). For excitation of Pyr-Tet-AF647 a 640-nm optically pumped semiconductor laser (OPSL) (Chroma, Genesis MX639-1000 STM, Coherent, Cleanup 640/10) was focused onto the back focal plane of the oil-immersion objective (Olympus, 60x, NA 1.45). Emission light was separated from the illumination light using a dichroic mirror (Semrock, FF 410/504/582/669 Brightline) and spectrally filtered by a bandpass filter (Semrock, 679/41 BrightLine HC). Images were recorded with an EMCCD (Andor, Ixon DU897). Resulting pixel size for data analysis was measured as 129 nm. For each dSTORM measurement, at least 15,000 frames at 50 Hz and irradiation intensities of ~2 kW cm$^{-2}$ were recorded by TIRF (total internal reflection fluorescence) illumination. Experiments were performed in PBS-based photoswitching buffer containing 100 mM β-mercaptoethylamine (MEA; Sigma-Aldrich) adjusted to pH 7.4. Image reconstruction was performed using rapidSTORM3.3[69]. Overview images were reconstructed with pixel size of 20 nm, whereas insets were calculated with 10 nm pixel size. Prior to dSTORM imaging, fluorescent image of XPH20::eGFP was acquired at 10 Hz using a 487 nm diode laser (TopticaPhotonics, iBEAM-SMART-488-S-HP), a dichroic mirror (Semrock, FF 410/504/582/669 Brightline), and a bandpass filter (Chroma, ET525/50).

**dSTORM imaging analysis.** Cluster analysis was conducted using a custom-written python script applying DBSCAN algorithm as well as Ripley K analysis on localization data in determined region of interests (ROIs). In advance, XPH20::eGFP images were merged in ImageJ (Fiji) with the corresponding super-resolved reconstructed image to identify synaptic and extrasynaptic areas. Contrast and brightness of eGFP signal was dilated using ImageJ to determine ROIs of similar size in neuronal spines for γ2 S44* and γ8 S72* (Supplementary Fig. 4b). Synaptic and extrasynaptic localization densities describe the number of localizations detected per ROI area. All dSTORM analysis was carried out on localizations in frames between 2000 and 15,000, with intensity of more than 6500 camera counts and with a local background of less than 800. DBSCAN (with parameter epsilon of 20 nm and minPoints of 3) was applied for identification of clustered localizations of TARPs. Distributions for localizations per cluster and cluster area of synaptic and extrasynaptic γ2 S44* as well as γ8 S72* were displayed by their probability density function. The cluster density (number of clusters per ROI area) was calculated for clusters with less and more than 100 localizations per cluster. Cluster analysis was performed on 5 neurons of γ2 S44* (three independent experiments) and γ8 S72* (four independent experiments) resulting in analysis of synaptic γ2 S44* ROIs ($n = 104$), synaptic γ8 S72* ROIs ($n = 102$), extrasynaptic γ2 S44* ROIs ($n = 50$) and extrasynaptic γ8 S72* ROIs ($n = 52$).

We calculated and displayed Ripley's H-function, a normalized Ripley's K-function, as previously described[70,71]. Computation was carried out for each ROI without edge correction. The averaged H-function was compared to H-functions and their 95% confidence intervals as computed from 100 simulated data sets with localizations distributed on the same ROIs (and identical number of localizations in each ROI) according to complete spatial randomness or a Neyman-Scott process. The Neyman-Scott clustering process has homogeneously distributed parent events with each parent having $n$ offspring events, where $n$ is Poisson distributed with mean 10, and with the offspring positions having a Gaussian offset with a standard deviation of 12 nm. The maximum of the H-function indicates a distance that is between cluster radius and diameter and thus provides an estimate for the average cluster size.

**Frequency domain-based fluorescence lifetime imaging-Förster resonance energy transfer measurements.** HEK293T cells plated at a density of 50,000–60,000 cells per cm$^{-2}$ on a pre-coated PLL 4-well Nunc™ Lab-Tek™ II chamber were co-transfected with PylRS/tRNA$^{Pyl}$ (166 ng), ncAA-tagged TARPs (166 ng) and SNAP-tagged GluA1 (166 ng), or PylRS/tRNA$^{Pyl}$ and tethered GluA1 SNAP396::γ2 S61* or GluA1 SNAP396::γ8 K102* in equal amounts (250 ng) using jetPRIME®. 250 μM TCO*A and 40 μM NBQX were added to the cells at the time of the transfection. After 48 h, cells were incubated with 1.5 μM H-Tet-Cy3 and 5

μM BG-AF488 diluted in TCO*A-free medium for 30 min at 37 °C. Cells were rinsed three times with HBSS.

Experiments were performed in HBSS at 37 °C using an incubator box with an air heater system (Life Imaging Services) installed on an inverted Leica DMI6000 B (Leica Microsystem) spinning disk microscope and using the LIFA frequency-domain lifetime attachment (Lambert Instruments) and the LI-FLIM software. Cells were imaged with an oil-immersion objective (Leica, HCX PL Apo 100x/NA 1.4 oil) using an appropriate GFP filter set. Cells were excited using a sinusoidally modulated 3W 477 nm light-emitting diode at 40 MHz under widefield illumination. Fluorescence emission was collected using an intensified CCD LI2CAM MD camera (Lambert Instruments, FAICM). Lifetimes were referenced to a 1 mg mL$^{-1}$ erythrosine B that was set at 0.086 ns[72]. The lifetime of the sample was determined from the fluorescence phase-shift between the sample and the reference from a set of 12 phase settings using the manufacturer's LI-FLIM software. All data are pulled measurements from a minimum of 20 cells per individual preparation. At least 20 cells in a minimum of three individual preparations were taken in consideration, except GluA1 SNAP396::γ2 S44*-AF488/Cy3 which are from two preparations.

**Statistics**. All electrophysiological recordings were analyzed with IGOR Pro 5 (WaveMetrics). Current amplitudes were measured with built-in tools, and $\tau_{des}$ was measured with exponential fit using a least-squares algorithm.

Statistical significance was calculated using GraphPad Prism 9. Statistical values are given as mean ± SD or SEM (as indicated); ***$p < 0.001$, **$p < 0.01$, *$p < 0.05$, n.s. specifies no significance. Box and violin plot indicate 25th to 75th percentiles, with median represented as a centre line, and mean represented as a cross. On the box plot, whiskers represent min to max values (Fig. 2e and Supplementary Fig. 3a) or 1.5 times the interquartile range (Fig. 5c, d and Supplementary Fig. 7b). Statistical significance for the levels of total γ8 between untransfected and γ8 S72* transfected neurons (Supplementary Fig.5), and TARPs spine vs extraspine ratios in dissociated neurons (Fig. 3g) were analyzed using a two-tailed unpaired Welch's t-test. For multiple sample comparisons within electrophysiology experiments, one-way ANOVA with a Fisher's Least Significant Difference multiple comparisons test was used. For multiple sample comparisons within the FRET experiments, Welch's ANOVA multiple comparisons test was used. Sample sizes and biological replicates are given in the figure legends.

**Protocol exchange**. The methods regarding OHSC and dissociated neurons GCE and click labeling can be accessed here https://doi.org/10.21203/rs.3.pex-1691/v1[73].

**Reporting summary**. Further information on research design is available in the Nature Research Reporting Summary linked to this article.

## Data availability
All data supporting the findings of this study are provided within the paper and its supplementary information. The raw microscopy data underlying the results will be made available upon request to the corresponding authors. Requests will be answered within a week. Source data are provided with this paper.

## Code availability
The code to perform cluster analysis will be made available for research and reproducibility purposes upon request by contacting the corresponding author. Requests will be answered within a week.

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

## Acknowledgements
We thank Edward Lemke and Gemma Estrada Girona (European Molecular Biology Laboratory, Heidelberg, Germany) for the gift of the pCMV tRNA$^{Pyl}$/NESPylRS$^{AF}$ plasmid and expert training on how to use it. We wish to thank AS Hafner and F Coussen for early experiments to develop and characterize the γ2 and γ8 antibodies, and M Sainlos for the early gift of the Xph plasmids. We are gratefull to Andrew Plested and Ljudmila Katchan for suggesting some of the insertion sites in AMPAR subunits for FRET experiments and sharing the associated reagent. We thank the Bordeaux Imaging Center, part of the FranceBioImaging national infrastructure (ANR-10-INBS-04-0) for support in microscopy and in particular C. Poujol for advice and discussions on FLIM-FRET imaging; C. Martin and the IINS in vivo facility for animal husbandry. We thank the IINS cell biology core facilities (LABEX BRAIN [ANR-10-LABX-43]) and in particular C. Breillat and E. Verdier for cell culture and plasmid production. This work was supported by funding from the Ministère de l'Enseignement Supérieur et de la Recherche to D.C., Centre National de la Recherche Scientifique (CNRS), ERC grant ADOS (339541) and DynSynMem (787340) to D.C., grants from the conseil Régional d'Aquitaine to D.C., the MSCA-ITN-ETN SYNDEGEN (675554) to D.C. and D.B.N., Fondation Recherche Médicale (FDT202001010840) to D.B.N.; A.K., G.B., and M.S. acknowledge funding by the Deutsche Forschungsgemeinschaft (DFG, project SA829/19-1), the European Regional Development Fund (EFRE project "Center for Personalized Molecular Immunotherapy"), and the European Research Council (Synergy Grant ULTRA-RESOLUTION). S.D. acknowledges funding by the Deutsche Forschungsgemeinschaft (DFG, project DO1257/4-1).

## Author contributions
D.B.N. performed FRET, patch-clamp, and imaging experiments, developed the strategy for GCE in neurons, developed part of the strategies for cDNA constructs and their production, prepared neuronal samples, performed corresponding data analysis and figure preparation, and co-wrote the MS. A.K. and G.B. developed strategies for site-specific ncAA incorporation by GCE, established click mutants, performed mammalian cell culture experiments, and performed *d*STORM imaging. A.K., G.B., and S.D. performed the cluster analysis of the *d*STORM data, V.P. co-developed and co-performed the single-cell electroporation in organotypic slices, N.R. developed part of the strategies for cDNA constructs and their production and supervised the neuronal primary culture production, N.C. developed part of the strategies for cDNA constructs and their production, D.P. developed the patch-clamp approach and contributed to the corresponding analysis, M.S. and D.C. co-supervised the study and co-wrote the MS. All authors read and corrected the MS.

## Competing interests
The authors declare no competing interests.
