## [Peer Review File · Nature Communications]

Reviewers' Comments:

Reviewer #1:

Remarks to the Author:

Bessa-Netto et al utilize recent technology to incorporate non-canonical amino acids into the critical AMPA receptor subunits TARP $\gamma 2$ and $\gamma 8$. They assess labeling using a thorough and rigorous series of validation steps, demonstrating both the success of the labeling and the innocuousness of the amino acid incorporation. They then demonstrate that the site may be hidden in $\gamma 2$ during typical interactions with AMPARs, which advertises the high potential of the strategy for tagging proteins particularly those where easy N- or C-terminal tags are not tolerated. Using the technique, they confirm the subcellular distribution of the two TARPs in both cell cultures and brain slices, and show that the two are heavily associated with AMPARs on the neuronal surface. All in all, the paper is a beautiful and carefully designed technical demonstration applying powerful tools to important proteins that have been problematic to image. They demonstrate a workflow that should provide a model for similar work in neurons labeling other proteins. Nevertheless, the paper strikes a bit of an awkward balance, as it not clear exactly how much they have advanced the techniques (rather than apply them), and the discoveries made with them are not particularly unexpected (though the advances are welcome). Some work on both aspects of the paper would improve it.

The array of use in both heterologous cells, neuron cultures, and brain slices is definitely impressive. However, readers will have difficulty determining if the paper is a technical advance over other work published from some of these authors, or simply very beautiful applications.

1. Were any tools or protocols (other than the TARP engineering itself) generated here? Is the efficiency increased over previous iterations, or were there specific approaches developed that are uniquely useful in neurons? A summary of these points would be beneficial.
2. The impact of the work would be much greater if the path forward for readers to utilize it were explicitly outlined. This is not a call to turn this into a protocols paper, but the technology is complex and a streamlined overview here would be helpful.
3. Given the presence of amber codons throughout the genome, isn't it quite surprising that labeling was not observed in cells not transfected with the mutant protein (as indicated starting line 152)? A massively increased signal in transfected neurons may also reflect severe overexpression that would not be desirable. Further, a concern for somebody following in their footsteps is to have an understanding of how strict the expression of the PyIRS has to be to prevent off-target termination of proteins or off-target labelling. As the extent of the basal labeling may limit utility of the approach, this should be explored in neurons and analyzed in some detail, perhaps with a dox time course.
4. The authors do not shy away from mentioning limitations of the approach, including the potential for endogenous amber labeling and the need for expressing the mutant protein. Routes to multicolor imaging would be good to touch on as well.
5. The authors claim that this will be better for quantitative imaging. I tend to believe this, but is this shown directly to be true for a protein that can be labeled with current technology, or just inferred because some proteins are not labelable?

The biological findings are interesting though not surprising and still need some additional support.

6. Does the addition of mCherry disrupt $\gamma 2$ and lead to its detection in Supp Fig 1? It would be surprising, but the authors should add the mCherry to the GFP-AMPA- $\gamma 2$ fusion protein.
7. It might be appropriate here to determine the overexpression level, particularly for $\gamma 2$ where an antibody is available, since it may be a concern in a few places besides point 3 above.
 - a. How much does overexpression contribute to revealing $\gamma 2$ in neurons?
 - b. The protein distribution in Fig 1e vs Sup Fig 1e is quite different. Was this due to a different promoter or an effect of the mCh?
8. Aside from confirming the differing distribution of the two TARP family members, the main biological conclusion is that all TARPs are saturated with AMPARs. This is a bit surprising, since it implies no available pool of TARPs during receptor trafficking. However, the implications of this finding are not discussed, and it is not clear whether there is conflicting data in the literature among the various (less effective) published approaches that are pointed out in the discussion. This leaves the conclusions feeling a bit thin.
 - a. On a related point of style: the discussion consists principally of one long paragraph. This makes

it difficult to determine the main points of interest that the work adds to the literature.

Reviewer #2:

Remarks to the Author:

The manuscript by Sauer, Choquet and co-workers describes a method that allows for bioorthogonal labeling of transmembrane proteins of live neurons via minimally perturbing means (i.e., through genetic encoding of a bioorthogonally applicable non-canonical, amino acid (ncAA) via amber-codon suppression). While the advantages, in particular, the minimal perturbation of bioorthogonal chemistry in combination with ncAA incorporation over other specific protein labeling means (e.g., fusion of fluorescent proteins, TAGs etc.) have been demonstrated previously in many instances, this work is among the pioneering ones that applies this methodology on the highly challenging platform of live neurons. The authors managed to address the problem of masked epitope labeling by carefully designed constructs suitable for site-specific labeling with tetrazine bearing fluorescent probes. Due to minimal perturbation the labeled proteins showed mainly unaffected activity and were suitable to study the distribution of TARPs and their effects on AMPARs. The experiments are carefully designed and addressed concerns related to GCE- (e.g., presence of natural amber codons) or overexpression-related problems. As such, I think it is a valuable methodological paper, which is definitely that of wide interest. Since its strength is the methodology describing the manipulation of live neurons, the experiments should be described in a way that it could be followed and reproduced by less-experienced researchers. Therefore, I suggest acceptance after addressing the minor concerns below. While evaluating this work I came through a somewhat related work by Nikic et al. on Biorxiv (<https://www.biorxiv.org/content/10.1101/2021.01.14.426692v1.full>). The authors should pay attention to cite this work properly (especially regarding the CRISPR approach), upon acceptance of both works.

1. The text is hard to follow at some points (partly due to the English). Since I consider this work a methodologically pioneering description, which most probably will be read by prospective followers, the authors should pay attention to help the readers guide through the text more easily.

2. TCO* is a cyclooctene, which is also used in click and release schemes due to beta-elimination of the appending 3-OH. Although some works report tetrazine-TCO* conjugates stable, more examples indicate the occurrence of elimination. In this particular work such an elimination would result in the loss of the label from the Lys residue. Why did the authors choose this particular cyclooctene? Why not 5-OH-TCO or a cyclooctyne, e.g., BCN? Have the authors observed loss of fluorescence in time? Please comment on these.

3. The authors call TCO*-lysine as TCO*A, which is a bit confusing for A is standing for Alanine. TCO*K (or TCO*Lys) would be more appropriate.

4. The text should be more explicit at some points e.g., line 111 ...overexpressed in cells (what cells? COS9 or neurons?); or in lines 139-140 and 169 please make it evident that cells were co-transfected with only one ncAA-tagged TARP at a time (and not both) – the reader can find all these out later or from the experimental descriptions, but it would be much easier if one should not go back and forth between respective parts.

5. same problem in lines 183-184: ...did not observe significant difference....relative to what? Probably they refer to similar decrease in lifetime as in the case of the previously discussed FRET system.

6. The authors should refine the technical aspects and make it more understandable for less experienced researchers.

Reviewer #3:

Remarks to the Author:

Choquet, Sauer and coworkers report in this work on combination of genetically encoded unnatural amino acids (UAA) with SPIEDAC labeling to visualize and study surface TARPs in living neurons. The authors first demonstrate that the 'classical' approach using antibodies and fluorescent proteins does not allow for efficient labeling/visualization of endogenous TARPs possibly due to engagement in binding with LBD masking the extracellular epitopes. To provide a solution to the problem, the authors employed here the amber suppression method in combination with click labeling. The position for insertion of the UAA was first optimized in HEK293T cells and was shown not to impair the functional properties of TARPs (based on FRET and patch-clamp experiments). A series of experiments with dissociated hippocampal neurons and organotypic hippocampal slice cultures revealed distinct localization of the $\gamma 2$ and $\gamma 8$ on the surface of neurons, which was finally corroborated by high resolution dSTORM experiments.

The presented work nicely demonstrates how the combination of GCE with bioorthogonal click labeling can contribute to answering 'problematic' biological questions. In particular, and as shown here, in a situation where traditional antibody-based approaches or fusion to fluorescent proteins does not provide the necessary resolution or is hampered by 'steric' reasons. The ability to perform high-end fluorescence microscopy studies in/on living systems in similar situation will clearly advance the field beyond neurobiology. The 'model' $\gamma 2/\gamma 8$ system studied here highlights the benefits of high-resolution imaging (different localization) and was therefore well chosen.

I do not have the necessary background in neurobiology nor am I an expert in high-end microscopy. However, I have good experience in GCE and especially in click labeling applications of the TCO/Tz reaction. Therefore, I provide my review from this perspective.

I did not find any substantial improvements in the labeling protocol or the particular reagents used throughout the study. Obviously, this was not the purpose of the work. In fact, I appreciate the use of commercially available reagents. This means that the protocols and methodology could be easily adopted by others to study different biological systems.

I find the work well performed and presented. Sufficient experimental details are included so that others should be able to reproduce the procedures. I have only few minor points.

1) Regarding the Tz labeling, I wonder if the authors did not have any problems with background staining in the experiments? I did not find any relevant comment in the manuscript. It is possible that the use of cell impermeable Tz-dyes for labeling extracellular targets is just fine, while intracellular labeling could be complicated by background signal?

2) The authors briefly state (line 145) that three different Tz dyes were investigated during the initial experiments on HEK293 T cells. If the data are available, I would appreciate one supplementary figure where the labeling with the three Tz dyes is compared (H-Tet-Cy3, H-Tet-Cy5 and Pyr-Tet-ATTO643). This could be of interest to the community using the Tz/TCO chemistry for other bioimaging applications and will make the manuscript potentially attractive to broader audience.

3) is there any possibility that the developed FRET experiments could be used to study the dynamics of the interaction? Such experiments are beyond the scope of this manuscript, but if so, short comment on the potential use along these lines could further substantiate these efforts.

Point-by-point response to reviewer comments Bessa-Neto et al.

We would like to thank the reviewers for their constructive and helpful criticism. In the following, we provide a point-by-point response to each reviewer comment.

Comments by reviewer #1

We thank this reviewer for his kind appreciation of our work (*“All in all, the paper is a beautiful and carefully designed technical demonstration applying powerful tools to important proteins that have been problematic to image”*).

Comment: *“They demonstrate a workflow that should provide a model for similar work in neurons labeling other proteins. Nevertheless, the paper strikes a bit of an awkward balance, as it not clear exactly how much they have advanced the techniques (rather than apply them), and the discoveries made with them are not particularly unexpected (though the advances are welcome). Some work on both aspects of the paper would improve it.”*

Reply: Altogether, we have worked to clarify the novelty in our MS regarding both the technical and the scientific aspects. In a nutshell, from the methodological point of view, our MS describes a new pipeline that allows for the first-time labelling proteins in live neurons by genetic code expansion. While each individual step of the pipeline is not new, its assembly and achievement are new. In addition, single-molecule localization microscopy of a neuronal protein (TARP associated to AMPAR) that cannot be labelled by any other technique is also new. From the scientific point of view, the subcellular distribution of TARPs could not be observed previously at the light microscopy or single molecule level, due to the impossibility to label them in live neurons. Hence our report is new. While there was a report of TARP sub-cellular distribution using electron microscopy, the limits inherent to EM (sensitivity, antigen accessibility) make our report scientifically new and relevant. It is interesting to note for example that Inamura et al. ((Inamura et al., 2006)) could only detect $\gamma 2$ and $\gamma 8$ perisynaptically with TEM, likely due to epitope masking. Given the recent increased interest in the differential role of TARP $\gamma 2$ and $\gamma 8$ in synapse function (e.g. (Yu et al., 2021; Zeng et al., 2019)), our finding of their differential distribution at the subcellular nanoscale is particularly relevant. Also, our new antibodies to extracellular loops of $\gamma 2$ and $\gamma 8$ allow us to establish that there are virtually no free surface $\gamma 2$ and $\gamma 8$, this is a new finding (see also answer to Q8 below). We have added paragraphs to better highlight the contributions of our MS to both the technical and scientific aspects (e.g. P3, L 75-78, P3-4 L97-101, P11 L351-356, P11, L367-373, P124 L382-384, P13, L407-409, P14 L443-449).

Q1. *“Were any tools or protocols (other than the TARP engineering itself) generated here? Is the efficiency increased over previous iterations, or were there specific approaches developed that are uniquely useful in neurons? A summary of these points would be beneficial.”*

Reply: We thank the reviewer for his/her comment and added the requested information in the revised version of our manuscript. The reagents we used are commercial and the enzyme for GCE is published. The cDNAs are new including the idea of using tet-on promoters for neuronal expression. All the pipeline of the experiment created for neurons (transfection methods, timings, concentrations, ratios, labelling methods, sites of insertion in part) are new. The protocols are all new. For GCE in slices the whole pipeline and protocols are completely new. The rounds of optimization of inducible vectors are new. Our specific contribution to GCE in neurons is now better highlighted in different paragraphs as listed above.

Q2. *“The impact of the work would be much greater if the path forward for readers to utilize it were explicitly outlined. This is not a call to turn this into a protocols paper, but the technology is complex and a streamlined overview here would be helpful.”*

Reply: As suggested by the reviewer and editor, we will upload a protocol and use the 'protocol exchange' if our MS is accepted.

Q3. *“Given the presence of amber codons throughout the genome, isn't it quite surprising that labeling was not observed in cells not transfected with the mutant protein (as indicated starting line 152)? A massively increased signal in transfected neurons may also reflect severe overexpression that would not be desirable. Further, a concern for somebody following in their footsteps is to have an understanding of how strict the expression of the PyIRS has to be to prevent off-target termination of proteins or off-target labelling. As the extent of the basal labeling may limit utility of the approach, this should be explored in neurons and analyzed in some detail, perhaps with a dox time course.”*

Reply: We thank the reviewer for these interesting remarks and suggestions.

Regarding the overexpression issue, we performed a whole new set of experiments to quantify precisely the level of overexpression using our anti- $\gamma 8$ c-term antibody and comparing the expression level in WT or $\gamma 8$ S72* transfected neurons. As presented in the new **supplementary Fig 5**, while a few neurons did display a marked increase in $\gamma 8$ expression level, overall, the average $\gamma 8$ expression level was not significantly different in WT and $\gamma 8$ S72* transfected neurons. This validates our observed distribution.

Regarding the absence of off target labelling, this is indeed our observation as our excellent signal-to-noise ratio indicates that if there is any off-target incorporation, it is below our detection level. The rationale for this observation is the following: while tRNA suppression of endogenous Amber codons may occur, the efficiency of tRNA suppression depends on several factors, in particular the environment of the Amber codon, i.e. nucleotides up- and downstream, its position in the gene, and competition with release factors ((de la Torre and Chin, 2021); (Bartoschek et al., 2021)). Moreover, the detection of such mutant at the plasma membrane would require a bypass of the ER protein quality-control checkpoints, the protein would need to belong to the type II transmembrane proteins, i.e. containing an extracellular C-terminus, as well as, require the proper insertion of such mutant at the plasma membrane (it is important to note here that we only label surface expressed proteins, and this likely explains our excellent specificity).

In our experimental conditions, no unspecific labeling ('basal' labeling) was detected in cells not expressing one of the ncAA-tagged TARPs as illustrated in figure 3. Cells co-expressing PyIRS/tRNA pair and wild-type $\gamma 2$ or $\gamma 8$ in the presence of TCO*A showed levels similar to the empty areas of the coverslip, therefore, no 'basal labeling due to off-target' was detected. That said, the 'massively increased signal in transfected neurons' exclusively shows the specificity of tetrazine labeling for $\gamma 2$ S44* or $\gamma 8$ S72*. We are aware that overexpression is of concern when studying protein-related functions, however, the current work represents a proof-of-concept towards the use of protein labeling via GCE in the context of neuroscience. Here we mainly focused on achieving a good balance between signal-to-noise ratio of tetrazine signal (based on TARPs expression levels). We kept in mind the problem of overexpression and that's why we used a doxy-inducible system, which was based on previous experience when working with TARPs. As mentioned above, our quantification of overexpression levels demonstrates that on average, expression levels of overexpressed $\gamma 8$ are not significantly different from endogenous levels (**supplementary Fig 5**). We extensively discuss this aspect in the revised version of our manuscript **P8 L266 to P9 L279**.

As for the expression PyIRS, it is difficult to say what would be the range of PyIRS levels to avoid any off-target effects or at levels that can be neglected, primarily as we did not observe any. Here the best approach would always be to restrict the expression of PyIRS and ncAA-tagged protein to the minimum amount of time required, which at the end, is always a good practice for overexpression systems. While we usually overexpressed for 20hrs, we also observed a decent expression (tetrazine labeling) at 16hrs. For the various reasons

described above, we respectfully believe that performing a dox time course as suggested is not necessary. Nevertheless, to further make this case clear, we now present experiments where we co-transfected dissociated neurons with a clickable GFP, Tet3G/tRNA^{Pyl} and pTRE3G-BI PylRS (the same system used for the expression of clickable TARPs in neurons), and treated with doxy+TCO*A for 5 days before H-Tet-Cy5 labeling (now presented as supplementary Figure 4). As shown by the line scans, no evident increase in H-Tet-Cy5 labeling is observed in transfected cells. Indeed, similar background fluorescence levels can be observed between transfected and non-transfected neurites (see H-Tet-Cy5 image, supplementary fig 4a bottom). Of note, a higher increase of H-Tet-Cy5 labeling was observed in the soma, however, this pattern was always observed in both non- and transfected cells. Together with the pre-requisites above mentioned for the detection of signal and the new data provided, we are confident that “off-target” labeling is not a concern in our experiments.

Q4. *“The authors do not shy away from mentioning limitations of the approach, including the potential for endogenous amber labeling and the need for expressing the mutant protein. Routes to multicolor imaging would be good to touch on as well.”*

Reply: We thank the reviewer for this suggestion and added a sentence in the discussion about potential routes to multicolour labelling with our small tags or bioorthogonal GCE P13 L427-437 and P14 L444-454.

Q5-1. *“The authors claim that this will be better for quantitative imaging. I tend to believe this, but is this shown directly to be true for a protein that can be labeled with current technology, or just inferred because some proteins are not labelable?”*

Reply: The main interest of the GCE technology, beyond labelling proteins with inaccessible epitopes, is the ability to perform perfectly controlled stoichiometric labelling. We have expanded a discussion point on this matter P14 L444-454. As TCO*-tetrazine labeling exhibits a ratio of 1, and each tetrazine is labeled with a single dye, this represents a great tool for quantitative studies. Other proteins as SNAP, HALO and FP also provide such modality, but their size ranges ~5 nm.

Q5-2. *“The biological findings are interesting though not surprising and still need some additional support.”*

Reply: We do not fully understand this comment nor the request. As mentioned above, our report is the first subcellular description of the differential $\gamma 2$ or $\gamma 8$ distribution using light and superresolution microscopy. The only two previous studies have used immunocytochemistry based EM approaches that have limitations in terms of sensitivity and epitope access. While the previous studies did already describe a differential distribution of $\gamma 2$ and $\gamma 8$, we respectfully believe that our study is more than a simple confirmation and is thus new enough to warrant publication (see also our reply to the comment above). We have expanded the discussion on the biological result novelty of our MS to answer the reviewer’s comment as listed in our reply in the first comment above.

Q6. *“Does the addition of mCherry disrupt $\gamma 2$ and lead to its detection in Supp Fig 1? It would be surprising, but the authors should add the mCherry to the GFP-AMPA- $\gamma 2$ fusion protein.”*

Reply: we respectfully believe that there is a misunderstanding of the reviewer and maybe a confusion with our previous paper (Chamma et al. 2016) where m-cherry was inserted in the extracellular loop and did disrupt $\gamma 2$. In this MS, the mCherry is at the C-term (before the important last c-ter PDZ domain binding ligand), and we have established previously that this type of construct is functional and can bind to PSD95 (e.g. Hafner et al. Neuron 2015). However, in order to homogenize Suppl. Fig 1e, we performed new experiments using the same c-terminal GFP tag in $\gamma 2$ and AMPAR- $\gamma 2$ fusion protein. Similar results were obtained,

i.e. correct expression of both constructs as visualized by GFP, but inability to antibody label $\gamma 2$ in the AMPAR- $\gamma 2$ fusion protein, demonstrating epitope masking.

Q7. *“It might be appropriate here to determine the overexpression level, particularly for $\gamma 2$ where an antibody is available, since it may be a concern in a few places besides point 3 above.”*

Reply: we thank the reviewer for this valuable suggestion. As listed above (Q3) we performed new experiments to measure the level of overexpression of $\gamma 8$. On average, the expression levels are not significantly different from endogenous levels. We used a $\gamma 8$ antibody for these experiments as we could not find a $\gamma 2$ antibody that provided sufficiently satisfactory staining. The $\gamma 2$ antibodies we tested displayed either no staining (extracellular epitope) or high background or could reveal only extrasynaptic gamma2 (c-term antibodies, likely due to masking of the c-term epitope that binds to PSD95). This is now extensively discussed in the revised version of our manuscript.

Q7a. *“How much does overexpression contribute to revealing $\gamma 2$ in neurons?”*

Reply: as mentioned above, we could not perform the analysis of the overexpression level of $\gamma 2$ due to a lack of an adequate antibody that can detect endogenous $\gamma 2$. In supplementary fig1e, the extracellular $\gamma 2$ antibody can only reveal the overexpressed $\gamma 2$ when not associated with AMPAR. The reviewer is correct in stating that overexpression of $\gamma 2$ can thus potentially contribute to revealing its labelling by antibody due to the presence of non-AMPAR associated $\gamma 2$. For this reason, we took great care to select low expressing cells in the GCE experiments. We are confident that the GCE approach reveals the correct distribution of $\gamma 2$ as we observed a strict synaptic localization, while strong overexpression would tend to saturate $\gamma 2$ antibody binding sites in the PSD and induce more extrasynaptic $\gamma 2$ that we did not detect. However, one should note that endogenous $\gamma 2$ is less expressed in hippocampal neurons than endogenous $\gamma 8$ and our overexpression approach probably leads to expression of $\gamma 2$ in neurons that would not normally express it, or only at low levels. This is a limitation inherent to the overexpression approach and we have introduced sentences in the discussion to raise this point. We respectfully believe this does not impair the main message of our paper.

Q7b. *“The protein distribution in Fig 1e vs Sup Fig 1e is quite different. Was this due to a different promoter or an effect of the mCh?”*

Reply: We respectfully believe that the reviewer got confused. There are two major differences between Fig1e and sup Fig 1e. First, in Fig1e the GFP is not fused to the TARP, contrary to sup Fig1e. Second, in Fig1e the staining of the TARP by GCE is only the surface TARP, while in sup Fig1e the mcherry signal (now the GFP signal in the revised figure, see point Q6 above) reveals surface and intracellular TARP. This explains the origin of the differences between Fig1e and sup Fig 1e. We’ve modified the text to make this clearer **P8 L255-257**.

Q8. *“Aside from confirming the differing distribution of the two TARP family members, the main biological conclusion is that all TARPs are saturated with AMPARs. This is a bit surprising, since it implies no available pool of TARPs during receptor trafficking. However, the implications of this finding are not discussed, and it is not clear whether there is conflicting data in the literature among the various (less effective) published approaches that are pointed out in the discussion. This leaves the conclusions feeling a bit thin. On a related point of style: the discussion consists principally of one long paragraph. This makes it difficult to determine the main points of interest that the work adds to the literature.”*

Reply: We thank the reviewer for these interesting remarks. The absence of detectable staining of endogenous $\gamma 2$ and $\gamma 8$ by the extracellular antibodies (that are otherwise functional) demonstrates that virtually all $\gamma 2$ and $\gamma 8$ extracellular loop epitopes are masked. As previous publications (Tomita et al., Science 2004) established that AMPAR are the only detectable $\gamma 2$ associated proteins, our data does indeed strongly suggest that all surface $\gamma 2$ associate to AMPAR (and by extension likely also for $\gamma 8$). We should mention that: 1) this is not a particularly controversial finding as it has not been previously specifically investigated to our knowledge, 2) this does not inform on the TARP association status in the biosynthetic pathway and our approach does not allow to answer this interesting question that has been studied using native gel shift for example in an interesting paper by the Fakler group (Schwenk, *et al.* Neuron 2019). In contrast, there is a controversy regarding the level of saturation of AMPAR with TARPs as each AMPAR can bear from zero to 4 TARPs with conflicting reports on the level of AMPAR saturation by TARPs. Our MS does not address this point. We have extended this point in the discussion (P 12, L382-384) and also separated it in several paragraphs as suggested to ease reading.

Comments by reviewer #2

We thank this reviewer for his kind appreciation of our work (“*..this work is among the pioneering ones that applies this methodology on the highly challenging platform of live neurons... I think it is a valuable methodological paper, which is definitely that of wide interest.*”).

Comment: “*...the experiments should be described in a way that it could be followed and reproduced by less-experienced researchers. Therefore, I suggest acceptance after addressing the minor concerns below. While evaluating this work I came through a somewhat related work by Nikic et al. on Biorxiv (<https://www.biorxiv.org/content/10.1101/2021.01.14.426692v1.full>). The authors should pay attention to cite this work properly (especially regarding the CRISPR approach), upon acceptance of both works.*”

Reply: We have added a reference to the nice pre-print of Nikic et al. (Arsić et al., 2021) as suggested (P 13, L430-437). We should however mention that the work of Arsic and colleagues represents not an overlap but rather a complementary tool to the current study. Regarding the CRISPR approach, we have to mention that the approach they use to incorporate the amber stop was through the CRISPR C-terminal-insertion of a mutant FLAG (N-terminal uAA-tagged FLAG tag; or clickable-GFP). Our approach focuses on site-specific insertion of a single uAA at TARPs extracellular loops, which is, currently, not easy to access through CRISPR technology as most of the available CRISPR strategies target the N- and C-terminus. Moreover, the use of a second tag to deliver the uAA via CRISPR, as in Arsic *et al.*, cancels the benefits of using GCE. We will aim in the future also using CRISPR to insert the Amber codon on the endogenous protein, but this is beyond our current MS. These elements are now inserted as discussion points P 13-14.

Q1. “*The text is hard to follow at some points (partly due to the English). Since I consider this work a methodologically pioneering description, which most probably will be read by prospective followers, the authors should pay attention to help the readers guide through the text more easily.*”

Reply: We thank the reviewer for his/her comment. We rephrased all respective paragraphs to improve the overall readability and comprehension.

Q2. “*TCO* is a cyclooctene, which is also used in click and release schemes due to beta-elimination of the appending 3-OH. Although some works report tetrazine-TCO* conjugates stable, more examples indicate the occurrence of elimination. In this particular work such an elimination would result in the loss of the label from the Lys residue. Why did the authors*

choose this particular cyclooctene? Why not 5-OH-TCO or a cyclooctyne, e.g., BCN? Have the authors observed loss of fluorescence in time?"

Reply: We thank the reviewer for raising this important point.

The use of either 3-OH or 5-OH is rarely discussed in detail when it comes to fluorescence labelling and is still an under-investigated subject in the field. Nevertheless, it was shown by the group of Edward Lemke, that the reactive cyclooctene at the position 3-OH (which they termed "TCO*") is at least ten times more stable than 5-OH ("TCO") or 4-OH ("TCO#") (Nikic et al., *Angewandte Chemie*, 2014). While all three derivatives showed similar reactivity, TCO* showed highest stability in the context of nucleophilic attack of other molecules like thiols. Additionally, TCO* and TCO# are approximately 3x better incorporated to the POI because of higher acceptance from the tRNA/PyIRS pair compared to TCO.

On the other side, the Robillard group has shown that TCO-derivatives can undergo β -elimination (Versteegen et al., *Angewandte Chemie*, 2013) caused by a post-click tautomerization of 3-OH functionalized TCO. While this enables great potential for drug-release and prodrug strategies, this might of course reduce label density and disturb stoichiometric imaging via SMLM. Notably, this effect is highly dependent on the side chains of the TCO-derivatives as well as pH- and buffer-dependent (Carlson et al., *JACS*, 2018)

Lastly, BCN represents an alternative to bypass the problem of elimination. However, strained alkynes (such as BCN or SCO) show much less reactivity to methyl-tetrazine (Me-Tet) than strained alkenes (such as TCO or NBO). Furthermore, BCN also reacts with azide-functionalized biomolecules, which can be advantageous if azide-dyes are desired, but would exclude a second click reaction exploiting (via SPAAC) besides the SPIEDAC reaction (between TCO*-lysine and a tetrazine-dye) enabled via genetic code expansion (GCE).

Overall, we anticipated that the readership and potential users would benefit most in using the present combination of TCO* and tetrazines, because it combines highest stability of the individual compounds, best incorporation efficiency as well as compatibility with other labelling approaches.

Q3. *"The authors call TCO*-lysine as TCO*A, which is a bit confusing for A is standing for Alanine. TCO*K (or TCO*Lys) would be more appropriate."*

Reply: TCO*A is the commercial name from SiChem for trans-Cyclooct-2-en – L – Lysine (SC-8008, SiChem, Bremen, Germany), where "A" stands for the "axial isomer". We thus do not feel legitimate to change the name. However, we better explained the name P3 L91 to help the reader.

Q4. *"The text should be more explicit at some points e.g., line 111 ...overexpressed in cells (what cells? COS9 or neurons?); or in lines 139-140 and 169 please make it evident that cells were co-transfected with only one ncAA-tagged TARP at a time (and not both) – the reader can find all these out later or from the experimental descriptions, but it would be much easier if one should not go back and forth between respective parts."*

Reply: we've modified the text accordingly

Q5. *"same problem in lines 183-184: ...did not observe significant difference....relative to what? Probably they refer to similar decrease in lifetime as in the case of the previously discussed FRET system."*

Reply: we've modified the text accordingly

Q6. *"The authors should refine the technical aspects and make it more understandable for less experienced researchers."*

Reply: We rephrased all respective paragraphs to improve the overall readability and comprehension. We will contribute a protocol exchange if our paper is accepted for publication as mentioned above.

Comments by reviewer #3

We thank this reviewer for his kind appreciation of our work (“*The presented work nicely demonstrates how the combination of GCE with bioorthogonal click labeling can contribute to answering ‘problematic’ biological questions...*” and “*I find the work well performed and presented.*”

Q1. “*Regarding the Tz labeling, I wonder if the authors did not have any problems with background staining in the experiments? I did not find any relevant comment in the manuscript. It is possible that the use of cell impermeable Tz-dyes for labeling extracellular targets is just fine, while intracellular labeling could be complicated by background signal?*”

Reply: The used tetrazine-derivatives are selected for optimal labeling efficiency with reduced background signals. This was investigated in our previous work ((Beliu et al., 2019)), where we characterized the click specificity as well as the turn-on and further photophysical properties. Nevertheless, we added the following sentence with the respective citation to the main text P11 L 360-363: “As some tetrazine-dyes tend to bind non-specifically to intracellular compartments, our data (not shown) and previous work show that it is important to carefully select suitable tetrazine-dyes and establish proper control experiments for each specific application³¹.”

In addition, as mentioned in our reply to Q3 of reviewer 1, we have ample evidence for an absence of detectable “off-target” labeling of surface proteins and now provide a new supplementary Figure 4 to further illustrate this point (see Q3 reviewer 1). We indeed found intracellular labeling with Tz less clean and did not explore it further. We now comment this point in the MS as data not shown.

Q2. “*The authors briefly state (line 145) that three different Tz dyes were investigated during the initial experiments on HEK293 T cells. If the data are available, I would appreciate one supplementary figure where the labeling with the three Tz dyes is compared (H-Tet-Cy3, H-Tet-Cy5 and Pyr-Tet-ATTO643). This could be of interest to the community using the Tz/TCO chemistry for other bioimaging applications and will make the manuscript potentially attractive to broader audience.*”

Reply: We thank the reviewer for this interesting suggestion. We added a new supplementary figure (Suppl. Figure 2), which compares the labeling performance of the three used tetrazine-dyes as well as negative controls.

Q3. “*is there any possibility that the developed FRET experiments could be used to study the dynamics of the interaction? Such experiments are beyond the scope of this manuscript, but if so, short comment on the potential use along these lines could further substantiate these efforts.*”

Reply: We thank the reviewer for this interesting suggestion. Indeed, we are planning to use this tool for the study of the dynamics of AMPAR and TARPs, in particular, in the context of synaptic plasticity. However, further development on the FRET pairs is required. We are currently working on new strategies to label AMPAR with smaller tags, like α -bungarotoxin binding site-tag, which we recently used in combination with GCE to study association/dissociation of heterodimers at the cell surface ((Beliu et al., 2021)). Probably one of the most controversial question about AMPAR and AMPAR-auxiliary proteins

interaction is whether AMPAR and, for example, TARPs can dissociate or not. In previous work, we observed that glutamate application induces a transient dissociation of $\gamma 2$ from AMPARs, however, this effect was abolished when $\gamma 2$ was overexpressed ((Constals et al., 2015)). Therefore, in addition to the use of smaller tags to label AMPARs, we are also working on strategies to combine GCE and CRISPR/Cas9 technology to achieve bioorthogonal labeling of endogenous TARPs. We have extended this point in the discussion.

- Arsić, A., Hagemann, C., Stajković, N., Schubert, T., and Nikić-Spiegel, I. (2021). Minimal genetically encoded tags for fluorescent protein labeling in living neurons. *bioRxiv*, 2021.2001.2014.426692.
- Bartoschek, M.D., Ugur, E., Nguyen, T.A., Rodschinka, G., Wierer, M., Lang, K., and Bultmann, S. (2021). Identification of permissive amber suppression sites for efficient non-canonical amino acid incorporation in mammalian cells. *Nucleic Acids Res* 49, e62.
- Beliu, G., Altrichter, S., Guixa-Gonzalez, R., Hemberger, M., Brauer, I., Dahse, A.K., Scholz, N., Wieduwild, R., Kuhlemann, A., Batebi, H., *et al.* (2021). Tethered agonist exposure in intact adhesion/class B2 GPCRs through intrinsic structural flexibility of the GAIN domain. *Mol Cell* 81, 905-921 e905.
- Beliu, G., Kurz, A.J., Kuhlemann, A.C., Behringer-Pliess, L., Meub, M., Wolf, N., Seibel, J., Shi, Z.D., Schnermann, M., Grimm, J.B., *et al.* (2019). Bioorthogonal labeling with tetrazine-dyes for super-resolution microscopy. *Communications biology* 2, 261.
- Constals, A., Penn, A.C., Compans, B., Toulme, E., Phillipat, A., Marais, S., Retailleau, N., Hafner, A.S., Coussen, F., Hosy, E., and Choquet, D. (2015). Glutamate-Induced AMPA Receptor Desensitization Increases Their Mobility and Modulates Short-Term Plasticity through Unbinding from Stargazin. *Neuron* 85, 787-803.
- de la Torre, D., and Chin, J.W. (2021). Reprogramming the genetic code. *Nature reviews. Genetics* 22, 169-184.
- Inamura, M., Itakura, M., Okamoto, H., Hoka, S., Mizoguchi, A., Fukazawa, Y., Shigemoto, R., Yamamori, S., and Takahashi, M. (2006). Differential localization and regulation of stargazin-like protein, gamma-8 and stargazin in the plasma membrane of hippocampal and cortical neurons. *Neurosci Res* 55, 45-53.
- Yu, J., Rao, P., Clark, S., Mitra, J., Ha, T., and Gouaux, E. (2021). Hippocampal AMPA receptor assemblies and mechanism of allosteric inhibition. *Nature* 594, 448-453.
- Zeng, M., Diaz-Alonso, J., Ye, F., Chen, X., Xu, J., Ji, Z., Nicoll, R.A., and Zhang, M. (2019). Phase Separation-Mediated TARP/MAGUK Complex Condensation and AMPA Receptor Synaptic Transmission. *Neuron* 104, 529-543 e526.

Reviewers' Comments:

Reviewer #1:

Remarks to the Author:

This is a thorough revision addressing all concerns. Congratulations to the authors on an interesting and surely difficult achievement.

Reviewer #2:

Remarks to the Author:

The authors have addressed my questions and concerns accordingly. The manuscript can now be accepted

Reviewer #3:

Remarks to the Author:

In the revised manuscript, the authors Choquet, Sauer and coworkers have addressed my original concerns. More specifically:

1) My first comment went to background labeling using different Tz-dyes. The authors provide additional experiments (Suppl. Fig. 4) which support their original claims and added comment to the main text. This addresses my concern.

2) Additional experiments showing labeling efficacy with three different Tz-dyes were added to Suppl. Fig. 2 as I suggested.

3) potential application of the developed FRET system to study the dynamics of the AMPAR/TARPs was added to the main text, so that this issue has been addressed as well.

I do not have additional comments or suggestions.

Point-by-point response to reviewer comments

Bessa-Neto et al.

We would like to thank the reviewers for their thorough review of our revised manuscript and their positive feed-back.

Comments by reviewer #1

“This is a thorough revision addressing all concerns. Congratulations to the authors on an interesting and surely difficult achievement.”

Reply: We are pleased that we have adequately addressed your concerns. Thank you very much for this positive evaluation.

Comments by reviewer #2

“The authors have addressed my questions and concerns accordingly. The manuscript can now be accepted”

Reply: We are pleased that we have adequately addressed your concerns. Thank you very much for this positive evaluation.

Comments by reviewer #3

“In the revised manuscript, the authors Choquet, Sauer and coworkers have addressed my original concerns. More specifically:

1) My first comment went to background labeling using different Tz-dyes. The authors provide additional experiments (Suppl. Fig. 4) which support their original claims and added comment to the main text. This addresses my concern.

2) Additional experiments showing labeling efficacy with three different Tz-dyes were added to Suppl. Fig. 2 as I suggested.

3) potential application of the developed FRET system to study the dynamics of the AMPAR/TARPs was added to the main text, so that this issue has been addressed as well. I do not have additional comments or suggestions.”

Reply: We are pleased that we have adequately addressed your concerns. Thank you very much for this positive evaluation.